# FedRot-LoRA: Mitigating Rotational Misalignment in Federated LoRA

**Haoran Zhang** [1]   **Dongjun Kim** [1]   **Seohyeon Cha** [1]   **Haris Vikalo** [1]

## Abstract

Federated LoRA provides a communication-efficient mechanism for fine-tuning large language models on decentralized data. In practice, however, a discrepancy between the factor-wise averaging used to preserve low rank and the mathematically correct aggregation of local updates can cause significant aggregation error and unstable training. We argue that a major source of this problem is *rotational misalignment*, arising from the rotational invariance of low-rank factorizations – semantically equivalent updates can be represented in different latent subspaces across clients since $(B_i R_i)(R_i^\top A_i) = B_i A_i$. When such misaligned factors are averaged directly, they interfere destructively and degrade the global update. To address this issue, we propose **FedRot-LoRA**, a federated LoRA framework that aligns client updates via orthogonal transformations prior to aggregation. This alignment preserves the semantic update while reducing cross-client subspace mismatch, without increasing communication cost or restricting model expressivity. We provide a convergence analysis that examines the aggregation error induced by factor-wise averaging and shows how rotational alignment yields a tighter upper bound on this error. Extensive experiments on natural language understanding and generative tasks demonstrate that FedRot-LoRA consistently outperforms existing federated LoRA baselines across a range of heterogeneity levels and LoRA ranks. The code is available at https://github.com/haoran-zh/FedRot-LoRA.

[1]Chandra Department of Electrical and Computer Engineering, The University of Texas at Austin, TX, USA. Correspondence to: Haoran Zhang <haoranz@austin.utexas.edu>.

*Proceedings of the 43rd International Conference on Machine Learning*, Seoul, South Korea. PMLR 306, 2026. Copyright 2026 by the author(s).

## 1. Introduction

The remarkable performance of large language models (LLMs) is driven by training and adaptation on large-scale, diverse datasets (Kaplan et al., 2020). In many real-world applications, however, high-quality data is distributed across devices or institutions and cannot be centralized due to privacy or regulatory constraints (Rieke et al., 2020). Federated learning (FL) (McMahan et al., 2017) offers a framework to leverage such decentralized data by coordinating training across clients while keeping raw data local.

In practice, federated fine-tuning faces major practical barriers (Zhang et al., 2023). Full-model training is often computationally infeasible on resource-constrained clients (Xu et al., 2024), and exchanging full model updates across training rounds introduces severe communication overhead. Parameter-efficient fine-tuning (PEFT) has emerged as an effective way to reduce computation and communication cost by updating only a small subset of parameters while keeping the backbone frozen (Lialin et al., 2023; Han et al., 2024). Among PEFT methods, Low-Rank Adaptation (LoRA) (Hu et al., 2022) parameterizes weight updates via low-rank factors ($\Delta W = BA$), making it particularly attractive in federated settings. Unlike prompt-based or adapter-based approaches, LoRA operates directly in the model weight space and incurs no additional inference-time overhead once the low-rank updates are merged, while remaining expressive under a low-rank assumption. These properties make LoRA well-suited for FL and have motivated a growing body of work on *federated LoRA*, which aims to combine FL's privacy guarantees and broad data coverage with LoRA's efficiency for practical decentralized LLM fine-tuning (Babakniya et al., 2023; Kuang et al., 2024).

**Challenge of Federated LoRA.** Integrating LoRA into FL introduces a fundamental aggregation challenge of balancing communication cost against mathematical correctness. Consider an FL system with $N$ clients, where each client $i$ in round $t$ computes a local update $\Delta W_i^t = B_i^t A_i^t$, with $B_i^t \in \mathbb{R}^{d \times r}$, $A_i^t \in \mathbb{R}^{r \times d}$, and $r \ll d$. The mathematically ideal aggregation at the server is

$$\Delta W_{ideal}^t = \frac{1}{N} \sum_{i=1}^{N} \Delta W_i^t = \frac{1}{N} \sum_{i=1}^{N} B_i^t A_i^t, \quad (1)$$

which provides the exact average of the local weight updates.

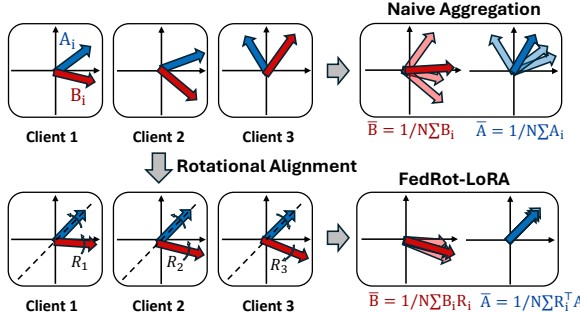

*Figure 1.* Overview of rotational alignment in FedRot-LoRA. Top: Naive aggregation averages unaligned LoRA factors $(A_i, B_i)$, causing destructive interference due to cross-client subspace mismatch. Bottom: FedRot-LoRA applies client-specific rotations $R_i$ to align local updates prior to aggregation. This preserves the semantic update while reducing subspace misalignment, leading to lower aggregation error and more stable training behavior.

However, $\Delta W_{ideal}^t$ is generally of rank greater than $r$, defeating the purpose of LoRA's low-rank parameterization and leading to prohibitive communication overhead when transmitting updates back to clients. Additionally, clients cannot directly resume LoRA training from a high-rank update, complicating continual fine-tuning. To preserve the low-rank structure, a commonly used alternative is to average the LoRA factors separately (Zhang et al., 2024a):

$$\Delta W_{naive}^t = \bar{B}^t \bar{A}^t = \left( \frac{1}{N} \sum_{i=1}^{N} B_i^t \right) \left( \frac{1}{N} \sum_{i=1}^{N} A_i^t \right). \quad (2)$$

While this guarantees $\mathrm{rank}(\Delta W_{naive}^t) \leq r$, enabling efficient communication and local re-initialization, it introduces a critical inconsistency ($\Delta W_{naive}^t \neq \Delta W_{ideal}^t$). This discrepancy manifests through cross terms $B_i^t A_j^t$ for $i \neq j$, which can destabilize training and degrade the performance of the global model (Chen et al., 2025; Bai et al., 2024).

**The Rotational Invariance Problem.** We argue that the limitations of naive factor-wise averaging cannot be fully explained by algebraic non-commutativity alone, but are strongly influenced by *rotational noise* arising from the rotational invariance of the LoRA decomposition. Given a local update $\Delta W_i^t = B_i^t A_i^t$, this factorization is not unique since for any orthogonal matrix $R \in \mathbb{R}^{r \times r}$, the transformed factors $\tilde{B}_i^t = B_i^t R$ and $\tilde{A}_i^t = R^\top A_i^t$ yield the same update, i.e., $\tilde{B}_i^t \tilde{A}_i^t = B_i^t A_i^t$. As a result, different clients may produce semantically identical updates while representing them in misaligned latent subspaces. When such unaligned factors are averaged directly, as in $\Delta W_{naive}^t$, combining mismatched subspaces leads to destructive interference, exacerbating aggregation error and adversely affecting training stability and global performance.

**Our Approach.** To mitigate the adverse effects of rotational misalignment, we propose **FedRot-LoRA**, a

communication-efficient federated fine-tuning framework that aligns latent subspaces of LoRA updates across clients prior to aggregation. Figure 1 illustrates the key idea. In each communication round, clients compute local low-rank updates that may span misaligned latent subspaces, leading to noisy aggregation and degraded global performance when averaged directly (top row of Fig. 1). To address this issue, FedRot-LoRA introduces a rotational alignment step that re-orients local LoRA factors before aggregation (bottom row of Fig. 1). This alignment preserves the semantic update while reducing cross-client subspace mismatch, mitigating rotational noise and improving training stability.

Our key contributions are summarized as follows:

- We identify *rotational noise*, induced by the rotational invariance of LoRA factorization, as an underexplored source of aggregation error in federated LoRA. To address this issue, we propose **FedRot-LoRA**, a federated fine-tuning framework that explicitly aligns local LoRA updates prior to aggregation, mitigating cross-client subspace mismatch under heterogeneous data.

- We provide a convergence analysis of federated LoRA that examines the aggregation error induced by factor-wise averaging, and show that rotational alignment yields a strictly tighter upper bound on this error in standard federated learning settings.

- Through extensive experiments on RoBERTa-Large (GLUE) and Llama 3-8B (GSM8K, HumanEval), we show that FedRot-LoRA consistently outperforms existing federated LoRA baselines across a wide range of client scales, LoRA ranks, and heterogeneity levels.

## 2. Related Work

### 2.1. Federated Parameter-Efficient Fine-Tuning

A growing body of work explores applying parameter-efficient fine-tuning (PEFT) methods in FL to adapt large models under resource constraints. **Prompt-based approaches** apply soft prompt tuning to avoid modifying the full model parameters. Examples include *FedPrompt* (Zhao et al., 2023) and *pFedPrompt* (Guo et al., 2023), which learn prompt representations that are either globally aggregated or personalized across clients. While these methods are highly communication-efficient and preserve model privacy, their adaptation capacity can be limited for tasks that require major changes to internal model representations (Lester et al., 2021). **Adapter-based approaches** introduce computationally efficient modules trained alongside the frozen backbone to capture task- or modality-specific information. $M^2$FedSA (Zhang et al., 2024b) incorporates specialized adapters to capture such information in federated settings. However, adapter-based methods incur additional inference

latency as adapters must be loaded and executed jointly with the backbone model at deployment. In contrast, LoRA-based methods (Wang et al., 2024b; Sun et al., 2024; Bai et al., 2024) compute updates via low-rank weight modifications that can be merged into the backbone after training, incurring no additional inference-time cost. Combined with their strong adaptation capacity under a low-rank assumption, LoRA-based approaches are particularly well-suited for federated fine-tuning of LLMs. Accordingly, in this work we focus on federated LoRA.

## 2.2. Federated LoRA

LoRA (Hu et al., 2022) reduces the parameter overhead of fine-tuning by reparameterizing weight updates as the product of two low-rank matrices. This proved to be an effective mechanism for adapting LLMs under limited computational and communication budgets (Hayou et al., 2024; Wang et al., 2024a). In FL settings, LoRA reduces the cost of fine-tuning across distributed clients by restricting communication to low-rank updates (Wang et al., 2024b). In practice, however, federated LoRA faces a key challenge: a mismatch between aggregating full local updates and aggregating low-rank factors, as in naive factor-wise averaging (Zhang et al., 2024a). Prior work has explored several strategies to mitigate this issue. **(1) Linearization via Parameter Freezing.** Methods such as *FFA-LoRA* (Sun et al., 2024) freeze one LoRA factor and aggregate only the other factor, ensuring linearity of aggregation. *RoLoRA* (Chen et al., 2025) alternates the frozen factor across training rounds. While these approaches simplify aggregation, they significantly restrict the parameter space, which may slow convergence and limit adaptation capacity. *FedSA-LoRA* (Guo et al., 2025) trains both factors locally but aggregates only a subset, leading to incomplete global adapters. **(2) Decomposition and Reconstruction.** Other methods perform aggregation in the full-weight space and then project the result to a low-rank form. For example, *FlexLoRA* (Bai et al., 2024) aggregates full updates and applies SVD-based projection to recover low-rank adapters, while *FedSRD* (Yan et al., 2025) reconstructs global updates from sparsified client updates before projection. These approaches incur substantial server-side computation and introduce approximation error due to repeated high-dimensional SVD operations, which can become numerically unstable (Chen et al., 2025). **(3) High-Communication Aggregation.** Another line of work preserves aggregation fidelity by transmitting extra information beyond low-rank adapters. *FedEx-LoRA* (Singhal et al., 2024) augments updates with high-rank residual terms, while *FLoRA* (Wang et al., 2024b) aggregates in the full parameter space with increasing effective rank. These methods substantially increase communication overhead, undermining the motivation for LoRA in bandwidth-constrained FL. We compare these federated LoRA methods with FedRot-

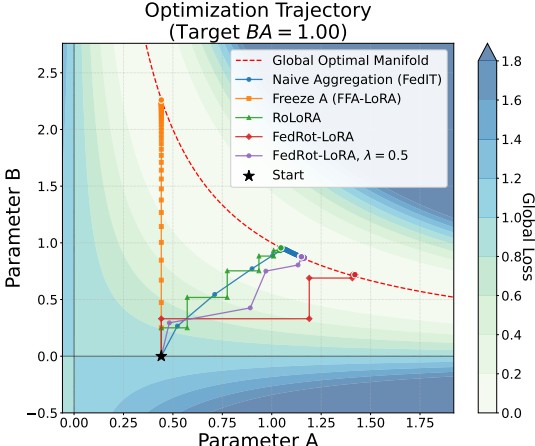

*Figure 2.* Optimization trajectories with target $\Delta W^* = 1.0$, initialized at $\bar{B}^0 = 0$ and $\bar{A}^0 = 0.44$ (black star). The red dashed curve denotes the optimal solution manifold. Each marker along a trajectory represents the global factors $(B, A)$ after one communication round under a given method. Different aggregation schemes induce distinct optimization paths, illustrating the impact of rotational misalignment on training dynamics.

LoRA in terms of aggregation space, communication, and computation in Table 7 of Appendix B.1.

In contrast to these methods, our work isolates rotational misalignment as a distinct contributor to aggregation error in federated LoRA and addresses it by explicitly aligning client updates prior to aggregation.

## 3. Methods

We consider an FL system with $N$ clients that jointly minimize the global objective $f(W) = \frac{1}{N} \sum_{i=1}^{N} f_i(W)$, where the model parameters are updated relying on a LoRA parameterization $W = W_0 + \Delta W = W_0 + BA$ with $B \in \mathbb{R}^{d \times r}, A \in \mathbb{R}^{r \times d}$ ($r \ll d$). In communication round $t$, client $i$ locally optimizes $f_i$ to obtain the low-rank update $\Delta W_i^t = B_i^t A_i^t$. Since direct aggregation of the corresponding LoRA factors can introduce a discrepancy between the aggregated update and the average of local updates, we introduce a rotational alignment step that reorients local LoRA factors prior to aggregation.

**Motivating Example.** We begin with a scalar example to illustrate the effect of aggregation misalignment. Consider the optimization problem with global loss $\ell(B, A) = (BA - 1)^2$, where $B, A \in \mathbb{R}$. The set of global minimizers satisfies $BA = 1$, forming a hyperbolic solution manifold (Fig. 2, red dashed line). Due to factorization invariance, different clients may converge to distinct but equivalent parameters (e.g., $B_i^t = 2, A_i^t = 0.5$ versus $B_j^t = 1.5, A_j^t = 2/3$). Naive factor averaging (FedIT (Zhang et al., 2024a), blue line) shifts the aggregate update off this manifold, introducing aggregation error and

unstable global updates. Parameter-freezing methods (FFA-LoRA (Sun et al., 2024), RoLoRA (Chen et al., 2025)) avoid this by aggregating a single factor, at the cost of reduced expressivity and slower convergence.

FedRot-LoRA instead aligns local factors prior to aggregation. In the scalar case, alignment reduces to rescaling: given a reference $A_{\text{ref}}$ (e.g., the previous global parameter $\bar{A}^{t-1}$), each client solves $\min_{R_i^t} \|R_i^t A_i^t - A_{\text{ref}}\|^2$ with $R_i^t \in \mathbb{R}$, and applies the transformation $(A_i^t, B_i^t) \rightarrow (R_i^t A_i^t, B_i^t / R_i^t)$ which preserves $B_i^t A_i^t$, yielding stable aggregated updates and faster convergence (red line). We further introduce a soft alignment mechanism (purple line) to improve robustness in early rounds. While the scalar case admits a simple rescaling interpretation, higher-rank LoRA ($r > 1$) is characterized by a more complex invariance setting where different clients may span misaligned latent subspaces. There, alignment generalizes from scalar rescaling to transformations that act on the latent subspace. Simple extensions such as diagonal or norm-based scaling cannot resolve relative subspace misalignment, while unrestricted invertible mappings may lead to ill-conditioned factors and require iterative, non–closed-form optimization. As a result, FedRot-LoRA adopts orthogonal alignment, which preserves invertibility, admits an efficient Procrustes solution, and provides $r(r-1)/2$ degrees of freedom for subspace alignment. A detailed comparison between rescaling and rotation is provided in Appendix B.3.1.

**Alternating Factor Alignment.** To mitigate subspace misalignment during aggregation, FedRot-LoRA introduces an alignment step prior to transmitting $B_i^t$ and $A_i^t$ to the server. We formulate this as an alternating optimization problem, where each client aligns its local LoRA factors to a common global reference, defined by the aggregated adapter from the previous round (i.e., $A_{\text{ref}} = \bar{A}^{t-1}$ and $B_{\text{ref}} = \bar{B}^{t-1}$).[1] To balance alignment across both factors, clients alternate between aligning $A$ and $B$ in consecutive communication rounds. Specifically, each client solves

$$\min_{R_i^t} \|(R_i^t)^\top A_i^t - A_{\text{ref}}\|_F^2, \qquad \text{if } t \bmod 2 = 1$$
$$\min_{R_i^t} \|B_i^t R_i^t - B_{\text{ref}}\|_F^2, \qquad \text{if } t \bmod 2 = 0$$
$$\text{s.t.} \quad (R_i^t)^\top R_i^t = I_r, \ \det(R_i^t) > 0. \qquad (3)$$

This optimization is an instance of the orthogonal Procrustes problem (Schönemann, 1966), which admits a closed-form solution obtained via singular value decomposition (SVD). Let $M$ denote the correlation matrix, defined as $M = A_{\text{ref}}(A_i^t)^\top$ in $A$-alignment rounds ($t \bmod 2 = 1$) and $M = (B_{\text{ref}})^\top B_i^t$ in $B$-alignment rounds ($t \bmod 2 = 0$). Let $M = U\Sigma V^\top$ be its SVD; the rotation matrix that mini-

---

[1] We use the previous global model to avoid extra communication overhead; alternatives are discussed in Appendix B.3.2.

mizes the above objective is then given by

$$R_i^{t,*} = V \cdot \text{diag}(1, ..., 1, \det(UV^\top)) \cdot U^\top. \qquad (4)$$

**Soft Rotation.** In the early stages of training, the global reference may be noisy or unrepresentative due to limited aggregation history and client heterogeneity. In such cases, the optimal alignment matrix $R_i^{t,*}$ can induce corrections that are unreasonably large, destabilizing training. To improve robustness, we introduce *soft rotation*, which interpolates between no alignment and exact Procrustes alignment. Specifically, we form an interpolated matrix

$$R' = (1 - \lambda)I + \lambda R_i^{t,*}, \qquad (5)$$

and project it by computing its SVD $R' = U\Sigma V^\top$ and setting $R_{i,\text{soft}}^t = V \text{diag}(1, \ldots, 1, \det(UV^\top)) U^\top$. The aligned factors are defined as

$$\tilde{A}_i^t = (R_{i,\text{soft}}^t)^\top A_i^t, \quad \tilde{B}_i^t = B_i^t R_{i,\text{soft}}^t. \qquad (6)$$

This transformation preserves the semantic update, since $\tilde{B}_i^t \tilde{A}_i^t = B_i^t A_i^t = \Delta W_i^t$. The interpolation factor $\lambda \in [0, 1]$ controls the strength of alignment. When $\lambda = 1$, soft rotation reduces to hard alignment; when $\lambda = 0$, no alignment is applied and FedRot-LoRA boils down to naive factor-wise aggregation. By reducing reliance on a noisy reference in early rounds, soft rotation prevents over-correction and leads to more stable training behavior.

The full training procedure for FedRot-LoRA is summarized as Algorithm 1. In each communication round $t$, the server broadcasts the current global LoRA adapters $\{\bar{A}^{t-1}, \bar{B}^{t-1}\}$ to all clients. Client $i$ initializes local training from these weights and updates its local factors $A_i^t$ and $B_i^t$ on private data. Following local training, clients apply rotational alignment to their updated factors; the aligned factors, $\tilde{A}_i^t$ and $\tilde{B}_i^t$, are then sent to the server, which aggregates them to generate the updated global adapters $\bar{A}^t$ and $\bar{B}^t$.

**Complexity Analysis.** FedRot-LoRA adds a small per-round computational cost that depends only on the LoRA rank. The alignment process involves two steps: *(1) Correlation matrix construction.* Each client computes the correlation matrix $M \in \mathbb{R}^{r \times r}$, which requires a matrix multiplication with time complexity $O(d \cdot r^2)$. *(2) SVD.* An SVD is then performed on the resulting $r \times r$ matrix $M$, which incurs a cost of $O(r^3)$. For the soft rotation, a similar SVD on an $r \times r$ matrix is required. Overall, the per-round complexity of the alignment step is $O(d \cdot r^2 + r^3)$. In contrast, decomposition-based methods such as FlexLoRA (Bai et al., 2024) require repeated decompositions or projections involving matrices of dimension $d$, leading to substantially higher computational cost when $d \gg r$. In practice, LoRA ranks are small (e.g., $r \in \{4, 8, 16\}$), and thus the alignment cost is modest relative to local training. Furthermore, the rotational alignment step introduces no additional communication overhead beyond standard federated LoRA.

**Algorithm 1** FedRot-LoRA

1: **Input:** $N$ clients, learning rate $\eta$, rank $r$, rounds $T$.
2: **Server Init:** Global parameters $W = W_0 + \Delta W_0$, $\Delta W_0 = \bar{B}^0 \bar{A}^0 = \mathbf{0}$.
3: **for** round $t = 1$ to $T$ **do**
4:     Server broadcasts $\{\bar{A}^{t-1}, \bar{B}^{t-1}\}$ to all clients.
5:     **for** each client $i \in \{1, \dots, N\}$ in parallel **do**
6:         **Local Training:** Update local $A_i^t, B_i^t$ via SGD starting from $\bar{A}^{t-1}, \bar{B}^{t-1}$.
7:         **Rotational Alignment:**
8:         **if** $t \bmod 2 = 1$ (align $A$) **then**
9:             Solve $\min_{R_i^t} \|(R_i^t)^\top A_i^t - A_{\text{ref}}\|_F^2$, s.t. (3).
10:         **else**
11:             Solve $\min_{R_i^t} \|B_i^t R_i^t - B_{\text{ref}}\|_F^2$, s.t. (3).
12:         **end if**
13:         Compute rotation matrix $R_{i,\text{soft}}^t$ via Eq. (5).
14:         Align $\tilde{A}_i^t = (R_{i,\text{soft}}^t)^\top A_i^t$, $\tilde{B}_i^t = B_i^t R_{i,\text{soft}}^t$.
15:         Send $\{\tilde{A}_i^t, \tilde{B}_i^t\}$ to server.
16:     **end for**
17:     **Aggregate:** $\bar{A}^t = \frac{1}{N} \sum_{i=1}^N \tilde{A}_i^t$, $\bar{B}^t = \frac{1}{N} \sum_{i=1}^N \tilde{B}_i^t$.
18: **end for**

## 4. Theoretical Analysis

In this section, we study how naive (i.e., factor-wise) aggregation in federated LoRA affects standard nonconvex convergence bounds. We explicitly characterize the aggregation error that arises from averaging low-rank factors instead of the resulting updates, and show that FedRot-LoRA reduces this error, yielding a strictly tighter bound.

### 4.1. Convergence Analysis

For the convergence analysis, we make standard smoothness and bounded gradient assumptions, along with a mild boundedness condition on local LoRA updates.

**Assumption 4.1** (L-smoothness). The global objective function $f(W)$ is differentiable and $L$-smooth: $f(Y) \leq f(X) + \langle \nabla f(X), Y - X \rangle + \frac{L}{2}\|Y - X\|_F^2$.

**Assumption 4.2** (Bounded Gradients). For each client $i$ and round $t$, let $\xi_i$ denote random local data samples. The stochastic gradients with respect to each parameter block $\theta \in \{A, B, W\}$ are unbiased and have bounded norm: $\mathbb{E}[\nabla_\theta f_i(W^t, \xi_i)] = \nabla_\theta f_i(W^t)$, $\|\nabla_\theta f_i(W^t)\|_F \leq G_\theta$.

**Assumption 4.3** (Bounded LoRA Norms). For each client $i$, the local LoRA update satisfies $\|B_i^t\|_F \cdot \|A_i^t\|_F \leq \tau$, $\forall i, t$.

Assumption 4.1 is a standard smoothness condition used to derive descent-based bounds in non-convex optimization. Assumption 4.2 ensures that stochastic gradients are uniformly bounded. Assumption 4.3 limits the scale ambiguity inherent to the LoRA factorization and implies that, in the fine-tuning regime, local low-rank updates remain

bounded and client heterogeneity is limited. We define the aggregation error as

$$E^t = \left(\frac{1}{N}\sum_{i=1}^N B_i^t\right)\left(\frac{1}{N}\sum_{i=1}^N A_i^t\right) - \frac{1}{N}\sum_{i=1}^N B_i^t A_i^t. \quad (7)$$

$E^t$ quantifies the discrepancy between the aggregation produced by factor-wise averaging and the ideal aggregation of local updates, and can be interpreted as an additive perturbation to the global model update at round $t$.

We present the convergence analysis in Theorem 4.4, with the proof detailed in Appendix A.1. Our analysis follows standard $L$-smooth nonconvex FL optimization to bound the per-round descent of the global objective (Wang et al., 2020), while explicitly isolating the aggregation error term $E^t$. This formulation explicitly quantifies how misaligned LoRA updates affect the convergence bound.

**Theorem 4.4** (Convergence Analysis). *Let Assumptions 4.1, 4.2, and 4.3 hold, and let $f^* = \inf_W f(W)$. For a learning rate $\eta$, the following stationarity bound holds for the aggregated global model $W^t = W_0 + \bar{B}^t \bar{A}^t$ over $T$ communication rounds:*

$$\min_{t \in \{0, \dots, T\}} \mathbb{E}\left[\|\nabla_A f(W^t)\|_F^2 + \|\nabla_B f(W^t)\|_F^2\right] \quad (8)$$

$$\leq \frac{f(W^0) - f^*}{\eta T} + \frac{3L\eta^2 + 2}{2T\eta} \sum_{t=1}^T \mathbb{E}\left[\frac{\|E^{t+1}\|_F^2}{\eta^2}\right] + \mathcal{O}(\eta).$$

*Interpretation:* The bound consists of three terms: the initial optimality gap, the accumulated aggregation error $\|E^{t+1}\|_F^2$, and a learning-rate-dependent term $\mathcal{O}(\eta)$. The aggregation error term quantifies the effect of naive factor-wise aggregation. Under the ideal aggregation, this term vanishes, resulting in a tighter convergence bound at the cost of rank expansion and increased communication cost. FedRot-LoRA reduces this error by aligning local updates prior to aggregation; in the next subsection, we formalize how such alignment leads to a strictly tighter aggregation error bound.

### 4.2. Rotational Alignment Yields Tighter Error Bound

We here analyze how rotational alignment affects the aggregation error $\|E^t\|_F$ introduced by factor-wise averaging. Under mild assumptions, we show that the proposed alignment step yields a strictly tighter upper bound on this error term, thereby strengthening the convergence bound derived in the previous subsection.

Let $\tilde{A}_i^t(\lambda)$ and $\tilde{B}_i^t(\lambda)$ denote the soft-aligned LoRA factors for client $i$ in round $t$, obtained with $\lambda \in [0, 1]$. We quantify cross-client misalignment via the dispersion defined as

$$\Phi(\lambda) = \begin{cases} \frac{1}{N}\sum_{i=1}^N \|\tilde{A}_i^t(\lambda) - A_{\text{ref}}\|_F^2, & \text{if } t \bmod 2 = 1 \\ \frac{1}{N}\sum_{i=1}^N \|\tilde{B}_i^t(\lambda) - B_{\text{ref}}\|_F^2, & \text{if } t \bmod 2 = 0. \end{cases}$$

Note that $\Phi(0)$ corresponds to the dispersion under factor-wise aggregation without rotational alignment. We define the relative alignment gain as

$$\alpha(\lambda) = 1 - \frac{\Phi(\lambda)}{\Phi(0)}. \tag{9}$$

**Assumption 4.5** (Linear Lower Envelope). There exists a constant $c_0 > 0$ such that $\alpha(\lambda) \geq c_0 \lambda$ for all $\lambda \in (0, 1]$.

**Assumption 4.6** (Non-IID Data Distribution). There exist $\delta_A, \delta_B > 0$ such that $\|A_i^t - A_{\text{ref}}\|_F \geq \delta_A$ and $\|B_i^t - B_{\text{ref}}\|_F \geq \delta_B$.

**Assumption 4.7** (Bounded Dispersion). There exist $\kappa > 0$ such that the optimal rotation matrices $R_i^{t,*}$ satisfy: $\|R_i^{t,*} - I\|_F \leq \kappa \|A_i^t - A_{\text{ref}}\|_F$ when aligning $A$, and $\|R_i^{t,*} - I\|_F \leq \kappa \|B_i^t - B_{\text{ref}}\|_F$ when aligning $B$.

Assumption 4.5 characterizes the effectiveness of rotational alignment as a function of the alignment strength by requiring the gain to be lower bounded. Assumption 4.6 captures non-vanishing client drift induced by data heterogeneity. Assumption 4.7 requires that rotational corrections remain controlled when local updates are close to the reference. We provide empirical diagnostics supporting these assumptions in Appendix B.3.5.

**Theorem 4.8** (Error Bound Analysis). *In a round $t$ where matrix $A$ is actively aligned, and under Assumptions 4.3, 4.5, 4.6, and 4.7, the aggregation error satisfies*

$$\|E_{aligned}^t\|_F \leq \underbrace{\frac{1}{N} \sum_{i=1}^{N} \left( \|\tilde{B}_i^t - B_{\text{ref}}\|_F^2 + \|\tilde{A}_i^t - A_{\text{ref}}\|_F^2 \right)}_{\text{Error bound with rotational alignment}}$$

$$\leq \frac{1}{N} \sum_{i=1}^{N} \bigg( \underbrace{\|B_i^t - B_{\text{ref}}\|_F^2 + \|A_i^t - A_{\text{ref}}\|_F^2}_{\text{Error bound without alignment}}$$

$$- \underbrace{\Gamma(\lambda) \cdot \|A_i^t - A_{\text{ref}}\|_F^2}_{\text{Alignment gain}} \bigg),$$

*where $\Gamma(\lambda) = \left( c_0 - \frac{4\sqrt{\tau}\kappa\eta G_B}{\delta_A} \right) \lambda - 4\kappa^2 \lambda^2 \tau$, which is positive for a non-trivial range of $\lambda$ under mild conditions. An analogous result holds when aligning matrix $B$.*

**Corollary 4.9** (Feasible $\lambda$ for Strict Improvement). *If the learning rate satisfies $\eta < \frac{c_0 \delta_A}{4\sqrt{\tau}\kappa G_B}$, then there exists a non-trivial range of soft rotation strengths $\lambda$ for which the alignment gain in Theorem 4.8 is strictly positive. In particular, if $0 < \lambda < \min\left\{1, \frac{c_0 \delta_A - 4\sqrt{\tau}\kappa\eta G_B}{4\kappa^2 \tau \delta_A}\right\}$, the aggregation error admits a strictly smaller upper bound than that obtained without alignment.*

*Interpretation.* Theorem 4.8 characterizes the effect of rotational alignment on the aggregation error through an explicit upper bound. Compared with naive factor averaging, rotational alignment introduces a $\lambda$-controlled *alignment gain*. Corollary 4.9 shows that there exists a non-trivial range of $0 < \lambda < \min\left\{1, \frac{c_0 \delta_A - 4\sqrt{\tau}\kappa\eta G_B}{4\kappa^2 \tau \delta_A}\right\}$ for which this gain is strictly positive, yielding a strictly tighter upper bound than naive aggregation under the same analysis. This provides a concrete guideline for selecting $\lambda$ to balance alignment strength and training stability. This range is a conservative sufficient condition arising from worst-case bounds, rather than a tight characterization of all effective choices of $\lambda$. In practice, as shown in our sensitivity experiments (see Section 5), FedRot-LoRA remains effective over a broader range of $\lambda$ values, suggesting that the theoretical condition captures the existence of a favorable alignment regime while the empirically useful range can be larger.

# 5. Experiments

In this section, we evaluate FedRot-LoRA against representative federated LoRA baselines on natural language understanding and generative tasks. We vary the number of clients, LoRA rank, and degree of data heterogeneity to examine performance across a range of federated settings.

## 5.1. Experimental Setup

**Datasets and Models.** For natural language understanding tasks, we use the pretrained RoBERTa-Large (Liu et al., 2019) model from the HuggingFace Transformers library (Wolf et al., 2020) and evaluate all methods on five datasets from GLUE (Wang et al., 2018): SST-2, QNLI, MNLI, QQP, and RTE. For generative tasks, we use Llama 3-8B (Grattafiori et al., 2024), evaluating mathematical reasoning on GSM8K (Cobbe et al., 2021) and code generation on HumanEval (Chen et al., 2021), with models trained on CodeSearchNet (Husain et al., 2019).

**Baselines.** We compare FedRot-LoRA against three federated LoRA baselines. *FedIT* (Zhang et al., 2024a) trains both LoRA factors $A$ and $B$ and aggregates them via naive averaging $(\bar{A}, \bar{B})$, preserving full expressivity but being susceptible to aggregation errors due to subspace misalignment. *FFA-LoRA* (Sun et al., 2024) freezes $A$ with random initialization and trains only $B$, allowing linear aggregation rule. *RoLoRA* (Chen et al., 2025) alternates between updating $B$ (with $A$ frozen) and updating $A$ (with $B$ frozen) across communication rounds. Both FFA-LoRA and RoLoRA avoid bilinear factor aggregation, but achieve that by restricting the effective parameter space, which can reduce expressivity and slow convergence.

**Implementation Details.** We implement all methods using the FederatedScope-LLM framework (Kuang et al., 2024). All experiments are conducted on RTX 6000 Ada and GH200 GPUs. To evaluate robustness across

*Table 1.* GLUE benchmark results comparing different federated LoRA methods under three client scales ($N \in \{3, 10, 50\}$), using $r = 4$ and Dirichlet concentration parameter $h = 0.5$. FedRot-LoRA achieves the highest average accuracy and shows improved stability across tasks.

| Clients Num | Methods | SST-2 | QNLI | QQP | RTE | MNLI | Avg. |
|---|---|---|---|---|---|---|---|
| 3 | FedIT (Zhang et al., 2024a) | $0.953 \pm 0.001$ | $0.926 \pm 0.002$ | $0.771 \pm 0.098$ | $0.840 \pm 0.013$ | $0.866 \pm 0.001$ | 0.8712 |
| | FlexLoRA (Bai et al., 2024) | $0.949 \pm 0.003$ | $0.910 \pm 0.007$ | $0.832 \pm 0.008$ | $0.813 \pm 0.015$ | $0.845 \pm 0.008$ | 0.8698 |
| | FFA-LoRA (Sun et al., 2024) | $0.768 \pm 0.056$ | $0.926 \pm 0.002$ | $0.838 \pm 0.002$ | $0.830 \pm 0.005$ | $0.862 \pm 0.001$ | 0.8448 |
| | RoLoRA (Chen et al., 2025) | $0.951 \pm 0.004$ | $0.917 \pm 0.011$ | $0.841 \pm 0.012$ | $0.854 \pm 0.005$ | $0.868 \pm 0.005$ | 0.8862 |
| | FedRot-LoRA (Ours) | $\mathbf{0.954 \pm 0.001}$ | $\mathbf{0.926 \pm 0.002}$ | $\mathbf{0.842 \pm 0.002}$ | $\mathbf{0.868 \pm 0.014}$ | $\mathbf{0.876 \pm 0.002}$ | **0.8932** |
| 10 | FedIT (Zhang et al., 2024a) | $0.949 \pm 0.002$ | $0.913 \pm 0.004$ | $0.849 \pm 0.000$ | $0.621 \pm 0.054$ | $0.849 \pm 0.002$ | 0.8362 |
| | FlexLoRA (Bai et al., 2024) | $0.922 \pm 0.009$ | $0.876 \pm 0.018$ | $0.801 \pm 0.032$ | $0.592 \pm 0.030$ | $0.792 \pm 0.011$ | 0.7966 |
| | FFA-LoRA (Sun et al., 2024) | $0.946 \pm 0.001$ | $0.894 \pm 0.008$ | $0.849 \pm 0.002$ | $0.600 \pm 0.024$ | $0.840 \pm 0.002$ | 0.8258 |
| | RoLoRA (Chen et al., 2025) | $0.956 \pm 0.001$ | $0.911 \pm 0.015$ | $0.863 \pm 0.001$ | $\mathbf{0.805 \pm 0.007}$ | $0.858 \pm 0.005$ | 0.8786 |
| | FedRot-LoRA (Ours) | $\mathbf{0.958 \pm 0.000}$ | $\mathbf{0.921 \pm 0.010}$ | $\mathbf{0.868 \pm 0.001}$ | $0.798 \pm 0.005$ | $\mathbf{0.864 \pm 0.001}$ | **0.8818** |
| 50 | FedIT (Zhang et al., 2024a) | $0.928 \pm 0.023$ | $0.803 \pm 0.005$ | $0.814 \pm 0.022$ | $0.622 \pm 0.051$ | $0.673 \pm 0.023$ | 0.7680 |
| | FlexLoRA (Bai et al., 2024) | $0.752 \pm 0.135$ | $0.733 \pm 0.121$ | $0.739 \pm 0.066$ | $0.632 \pm 0.059$ | $0.677 \pm 0.032$ | 0.7066 |
| | FFA-LoRA (Sun et al., 2024) | $0.933 \pm 0.015$ | $0.851 \pm 0.004$ | $0.716 \pm 0.018$ | $0.658 \pm 0.038$ | $0.701 \pm 0.008$ | 0.7718 |
| | RoLoRA (Chen et al., 2025) | $0.851 \pm 0.176$ | $0.893 \pm 0.042$ | $0.847 \pm 0.003$ | $0.733 \pm 0.064$ | $0.798 \pm 0.114$ | 0.8244 |
| | FedRot-LoRA (Ours) | $\mathbf{0.947 \pm 0.004}$ | $\mathbf{0.920 \pm 0.012}$ | $\mathbf{0.854 \pm 0.002}$ | $\mathbf{0.787 \pm 0.004}$ | $\mathbf{0.859 \pm 0.003}$ | **0.8734** |

*Table 2.* FedRot-LoRA achieves an order-of-magnitude reduction in aggregation error compared to FedIT.

| Methods | SST-2 | QNLI | QQP | RTE | MNLI |
|---|---|---|---|---|---|
| FedIT | 7.49e-3 | 1.98e-3 | 1.04e-2 | 1.91e-3 | 3.98e-3 |
| FedRot-LoRA | 4.72e-4 | 1.34e-4 | 8.70e-4 | 2.11e-4 | 1.48e-4 |

runs, experiments are repeated with three random seeds, and we report the mean accuracy with standard deviation. We perform a grid search over the learning rate $\eta \in \{5e\text{-}4, 1e\text{-}3, 5e\text{-}3, 2e\text{-}2\}$ and the alignment strength $\lambda \in \{0.2, 0.4, 0.6, 0.8, 1.0\}$. For each method, hyperparameters are tuned separately using validation, and the best-performing configuration is reported in Appendix B.2. We set the number of local epochs to 20 and the number of communication rounds to 250 for all natural language understanding tasks, and to 30 local epochs and 200 communication rounds for generative tasks.

### 5.2. Language Understanding Tasks

Table 1 reports RoBERTa-Large performance on five GLUE natural language understanding tasks (SST-2, QNLI, MNLI, QQP, RTE), where we report mean test accuracy and standard deviation over three random seeds. All methods use LoRA rank $r = 4$, and we adopt the non-IID data partition scheme from (Li et al., 2022) using a Dirichlet distribution with concentration parameter $h = 0.5$. We consider three FL scales, with $N \in \{3, 10, 50\}$ clients. Under $N = 3$, FedRot-LoRA achieves the highest average accuracy (0.8932) among all methods. This improvement is consistent with the substantial reduction in aggregation error achieved by FedRot-LoRA (Table 2). The performance gains are more pronounced on challenging tasks such as MNLI and RTE, where limited data and label noise can amplify aggregation mismatch across clients. In contrast, for relatively easy binary tasks such as SST-2 and QNLI,

multiple methods approach similar performance ceilings, consistent with reduced sensitivity to aggregation effects in these settings. When scaling to $N = 10$, most baselines degrade, particularly on small-data tasks such as RTE, reflecting the increased difficulty of aggregating heterogeneous client updates. FedRot-LoRA remains the most accurate method on average (0.8818) and improves training stability, reducing the averaged standard deviation across tasks to 0.0034 compared to 0.0058 for the strongest baseline RoLoRA. At the larger scale of $N = 50$, the advantage of FedRot-LoRA becomes more pronounced: it achieves the best performance on all five tasks and improves the average accuracy to 0.8734, compared with 0.8244 for RoLoRA and 0.7680 for FedIT. This suggests that rotational alignment becomes increasingly beneficial as client heterogeneity and aggregation difficulty grow.

### (1) Effect of Rank.

Table 3 reports MNLI accuracy for different LoRA ranks $r \in \{2, 4, 8, 16, 24\}$ under a non-IID partition with Dirichlet concentration parameter $h = 0.5$ and $N = 3$ clients. FedRot-LoRA achieves the best performance across all ranks, with consistently low standard deviation. At the low-rank setting $r = 2$, FedRot-LoRA still performs strongly, indicating that rotational alignment does not hurt optimization even when the adaptation subspace is highly constrained. As the rank increases, aggregation becomes more challenging because clients may span higher-dimensional and more misaligned latent subspaces. This particularly affects RoLoRA, whose performance drops sharply at $r = 16$ and $r = 24$, with large variance across runs. In contrast, FedRot-LoRA remains stable at both moderate and high ranks, achieving 0.865 at $r = 16$ and 0.872 at $r = 24$ with very small standard deviation. These results suggest that FedRot-LoRA is robust across both capacity-limited low-rank settings and higher-rank settings.

*Table 3.* MNLI accuracy under different LoRA ranks ($r \in \{2, 4, 8, 16, 24\}$). FedRot-LoRA outperforms all baselines across ranks, demonstrating stable performance across ranks.

| Rank | Methods | ACC±std@MNLI |
|---|---|---|
| 2 | FedIT (Zhang et al., 2024a) | $0.844 \pm 0.051$ |
|  | FlexLoRA (Bai et al., 2024) | $0.811 \pm 0.108$ |
|  | FFA-LoRA (Sun et al., 2024) | $0.837 \pm 0.008$ |
|  | RoLoRA (Chen et al., 2025) | $0.860 \pm 0.002$ |
|  | FedRot-LoRA (Ours) | $\mathbf{0.877 \pm 0.005}$ |
| 4 | FedIT (Zhang et al., 2024a) | $0.866 \pm 0.001$ |
|  | FlexLoRA (Bai et al., 2024) | $0.845 \pm 0.008$ |
|  | FFA-LoRA (Sun et al., 2024) | $0.862 \pm 0.001$ |
|  | RoLoRA (Chen et al., 2025) | $0.868 \pm 0.005$ |
|  | FedRot-LoRA (Ours) | $\mathbf{0.876 \pm 0.002}$ |
| 8 | FedIT (Zhang et al., 2024a) | $0.861 \pm 0.001$ |
|  | FlexLoRA (Bai et al., 2024) | $0.836 \pm 0.021$ |
|  | FFA-LoRA (Sun et al., 2024) | $0.864 \pm 0.001$ |
|  | RoLoRA (Chen et al., 2025) | $0.869 \pm 0.004$ |
|  | FedRot-LoRA (Ours) | $\mathbf{0.877 \pm 0.002}$ |
| 16 | FedIT (Zhang et al., 2024a) | $0.855 \pm 0.003$ |
|  | FlexLoRA (Bai et al., 2024) | $0.852 \pm 0.010$ |
|  | FFA-LoRA (Sun et al., 2024) | $0.834 \pm 0.009$ |
|  | RoLoRA (Chen et al., 2025) | $0.722 \pm 0.215$ |
|  | FedRot-LoRA (Ours) | $\mathbf{0.865 \pm 0.002}$ |
| 24 | FedIT (Zhang et al., 2024a) | $0.857 \pm 0.001$ |
|  | FlexLoRA (Bai et al., 2024) | $0.864 \pm 0.009$ |
|  | FFA-LoRA (Sun et al., 2024) | $0.844 \pm 0.001$ |
|  | RoLoRA (Chen et al., 2025) | $0.746 \pm 0.274$ |
|  | FedRot-LoRA (Ours) | $\mathbf{0.872 \pm 0.001}$ |

**(2) Effect of Data Heterogeneity.** Table 4 reports MNLI performance under varying levels of data heterogeneity, controlled by the Dirichlet concentration parameter $h \in \{100, 1, 0.5\}$; larger values correspond to more uniform (IID) client data, smaller values induce stronger non-IID skew. In the near-IID setting ($h = 100$), FedRot-LoRA achieves the highest accuracy (88.4%) with low standard deviation, outperforming all baselines. Most methods perform relatively well in this regime, as aggregation is less sensitive to client misalignment in this setting, though FedRot-LoRA retains a consistent advantage. As heterogeneity increases ($h = 1$ and $0.5$), performance degrades across all methods. FedRot-LoRA maintains the strongest performance in all settings and exhibits smaller standard deviation compared to competing approaches, indicating improved robustness to client drift under increasingly non-IID conditions.

## 5.3. Natural Language Generation

We evaluate natural language generation on GSM8K (mathematical reasoning) and HumanEval (code generation, trained on CodeSearchNet) using Llama 3-8B. Table 5 reports the final performance metrics (exact match accuracy for GSM8K and pass@1 for HumanEval). FedRot-LoRA achieves the strongest performance on both tasks, reaching

*Table 4.* MNLI accuracy under varying levels of data heterogeneity ($h \in \{100, 1, 0.5\}$). FedRot-LoRA outperforms all baselines across settings, highlighting its robustness to both homogeneous and heterogeneous data distributions.

| Methods | $h = 100$ | $h = 1$ | $h = 0.5$ |
|---|---|---|---|
| FedIT | $0.868 \pm 0.003$ | $0.864 \pm 0.015$ | $0.866 \pm 0.001$ |
| FlexLoRA | $0.871 \pm 0.001$ | $0.839 \pm 0.017$ | $0.845 \pm 0.008$ |
| FFA-LoRA | $0.869 \pm 0.000$ | $0.856 \pm 0.003$ | $0.862 \pm 0.001$ |
| RoLoRA | $0.767 \pm 0.158$ | $0.780 \pm 0.140$ | $0.868 \pm 0.005$ |
| FedRot-LoRA | $\mathbf{0.884 \pm 0.001}$ | $\mathbf{0.873 \pm 0.001}$ | $\mathbf{0.876 \pm 0.002}$ |

*Table 5.* Performance on generative tasks. We report pass@1 scores for code generation (evaluated on HumanEval) and exact match accuracy for mathematical reasoning (evaluated on GSM8K). FedRot-LoRA outperforms all baselines on both tasks.

| Methods | HumanEval (pass@1) | GSM8K (acc) |
|---|---|---|
| FedIT (Zhang et al., 2024a) | $0.2877 \pm 0.017$ | $0.4293 \pm 0.018$ |
| FlexLoRA (Bai et al., 2024) | $0.2930 \pm 0.048$ | $0.3515 \pm 0.126$ |
| FFA-LoRA (Sun et al., 2024) | $0.3851 \pm 0.009$ | $0.4361 \pm 0.001$ |
| RoLoRA (Chen et al., 2025) | $0.2951 \pm 0.081$ | $0.3444 \pm 0.172$ |
| FedRot-LoRA (Ours) | $\mathbf{0.4088 \pm 0.012}$ | $\mathbf{0.4437 \pm 0.009}$ |

44.37% on GSM8K and 40.88% on HumanEval, consistently outperforming all baselines. RoLoRA exhibits substantially higher variance across runs, indicating reduced training stability in these generative settings. In contrast, FedRot-LoRA maintains lower standard deviation while achieving higher accuracy, suggesting improved robustness under federated fine-tuning for complex generative tasks. Overall, these results indicate that FedRot-LoRA extends effectively beyond classification to generative language tasks.

## 5.4. Ablation Study

**(1) With vs. Without Rotational Alignment.** FedIT serves as a non-rotational baseline in which clients aggregate LoRA factors via naive factor-wise averaging. As shown in Tables 1-5, FedRot-LoRA outperforms FedIT across all tasks and LoRA ranks (higher accuracy, lower standard deviation), indicating the benefit of aligning latent subspaces prior to aggregation. We further evaluate a *random rotation* baseline, in which clients apply arbitrary orthogonal transformations before aggregation. This variant achieves an MNLI accuracy of 0.3181 (see Appendix B.3.3), demonstrating that meaningful, rather than random, alignment is required.

**(2) Effect of Soft Rotation Level $\lambda$.** We evaluate FedRot-LoRA with $\lambda \in \{0, 0.1, 0.2, \ldots, 0.9, 1.0\}$ on MNLI and QQP (Fig. 3). Across both tasks, a wide range of $\lambda$ values outperforms the average performance of the baseline methods. Performance is highest for intermediate values of $\lambda$, while too small or too large values lead to reduced accuracy. On MNLI, performance degrades when $\lambda$ is set too high (e.g., $\lambda = 1.0$), suggesting that aggressive alignment can be detrimental when the global reference is noisy, particularly in early rounds. A similar trend is observed on QQP, where excessive alignment also leads to lower accuracy. These

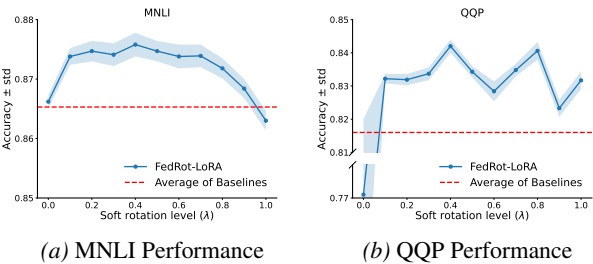

*(a)* MNLI Performance      *(b)* QQP Performance

*Figure 3.* Effect of soft rotation level $\lambda$ in FedRot-LoRA on MNLI and QQP. Red dashed line denotes the baseline's average accuracy.

*Table 6.* Ablation comparing alternating and single-factor alignment. FedRot-LoRA performs best, while single-factor alignment degrades accuracy, especially when aligning $B$ alone.

| Methods | SST-2 | QNLI | MNLI |
|---|---|---|---|
| Align A Only | $0.883 \pm 0.015$ | $0.913 \pm 0.008$ | $0.861 \pm 0.001$ |
| Align B Only | $0.879 \pm 0.015$ | $0.806 \pm 0.149$ | $0.862 \pm 0.017$ |
| FedRot-LoRA | $\mathbf{0.954 \pm 0.001}$ | $\mathbf{0.926 \pm 0.002}$ | $\mathbf{0.876 \pm 0.002}$ |

results indicate that soft rotation provides an effective way to control alignment strength, enabling gradual subspace alignment without destabilizing training.

**(3) Effect of Alternating Rotation Targets.** To test the importance of aligning both LoRA factors, we compare FedRot-LoRA with two variants that apply rotational alignment only to $A$ or only to $B$ throughout training. As shown in Table 6, both variants underperform the full method. The performance drop is most pronounced when aligning only $B$. One possible explanation is that $B$ is initialized with a small norm, which may limit the effectiveness of alignment in early training stages. In contrast, alternating the alignment target between $A$ and $B$ consistently yields better performance. These results indicate that alternating alignment across both LoRA factors is beneficial, and that restricting alignment to a single factor degrades performance.

Additional experimental results are deferred to Appendix B.

## 6. Conclusion

We identify rotational noise, caused by the rotational invariance of low-rank factorization, as a key source of aggregation error in federated LoRA. To address this issue, we propose FedRot-LoRA, which aligns local LoRA updates via client-side rotations prior to aggregation while preserving semantic equivalence. We provide a theoretical analysis showing that rotational alignment yields a strictly tighter bound on aggregation error induced by factor-wise averaging. Extensive experiments show that FedRot-LoRA outperforms existing federated LoRA methods across different client scales, ranks, and data heterogeneity levels.

## Acknowledgements

This work was funded in part by National Science Foundation (NSF) Grant 2148224.

## Impact Statement

This work improves federated fine-tuning of large language models by mitigating aggregation misalignment through a lightweight rotational alignment mechanism. This can facilitate the practical deployment of federated parameter-efficient fine-tuning in privacy-sensitive and resource-constrained settings. The work improves aggregation fidelity and training stability in federated learning and does not introduce additional societal risks beyond those inherent to federated learning and large language models.

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

# A. Proofs

## A.1. Proof of Theorem 4.4

*Proof.* We analyze the convergence of the global model $W^t$ under the proposed FedRot-LoRA. We denote the aggregated LoRA matrices at round $t$ as $\bar{A}^t = \frac{1}{N} \sum_{i=1}^{N} A_i^t$ and $\bar{B}^t = \frac{1}{N} \sum_{i=1}^{N} B_i^t$. The global model is constructed as $W^t = W_0 + \bar{B}^t \bar{A}^t$.

### 1. Definitions and Update Rules

Let the "ideal" global model update $W_{\text{ideal}}^{t+1}$ be the result of averaging the local weight updates directly, whereas the actual global model $W^{t+1}$ is formed by multiplying the averaged LoRA matrices.

$$W_{\text{ideal}}^{t+1} = W_0 + \frac{1}{N} \sum_{i=1}^{N} B_i^{t+1} A_i^{t+1} \tag{10}$$

$$W^{t+1} = W_0 + \left( \frac{1}{N} \sum_{i=1}^{N} B_i^{t+1} \right) \left( \frac{1}{N} \sum_{i=1}^{N} A_i^{t+1} \right) \tag{11}$$

We define the aggregation error $E^{t+1}$ as the difference between the actual and ideal updates:

$$E^{t+1} = W^{t+1} - W_{\text{ideal}}^{t+1} = -\frac{1}{2N^2} \sum_{i=1}^{N} \sum_{j=1}^{N} (B_i^{t+1} - B_j^{t+1})(A_i^{t+1} - A_j^{t+1}) \tag{12}$$

The global updates for the LoRA adapters follow the gradient descent rule with learning rate $\eta$.

### 2. Smoothness Analysis via Intermediate Step

Since the objective function $f$ is $L$-smooth, we have:

$$f(Y) \leq f(X) + \langle \nabla f(X), Y - X \rangle + \frac{L}{2} \|Y - X\|^2 \tag{13}$$

where $\| \cdot \|$ indicates Frobenius norm. We decompose the transition from $W^t$ to $W^{t+1}$ into two steps: $W^t \to W_{\text{ideal}}^{t+1}$ and $W_{\text{ideal}}^{t+1} \to W^{t+1}$.

*Step 2a: Gap between $W^{t+1}$ and $W_{\text{ideal}}^{t+1}$*
Applying smoothness between $W^{t+1}$ and $W_{\text{ideal}}^{t+1}$:

$$f(W^{t+1}) \leq f(W_{\text{ideal}}^{t+1}) + \langle \nabla f(W_{\text{ideal}}^{t+1}), W^{t+1} - W_{\text{ideal}}^{t+1} \rangle + \frac{L}{2} \|W^{t+1} - W_{\text{ideal}}^{t+1}\|^2$$

$$= f(W_{\text{ideal}}^{t+1}) + \langle \nabla f(W_{\text{ideal}}^{t+1}), E^{t+1} \rangle + \frac{L}{2} \|E^{t+1}\|^2 \tag{14}$$

Using Young's Inequality $\langle x, y \rangle \leq \frac{1}{2\eta^2} \|x\|^2 + \frac{\eta^2}{2} \|y\|^2$ on the inner product term:

$$\langle E^{t+1}, \nabla f(W_{\text{ideal}}^{t+1}) \rangle \leq \frac{1}{2\eta^2} \|E^{t+1}\|^2 + \frac{\eta^2}{2} \|\nabla f(W_{\text{ideal}}^{t+1})\|^2 \leq \frac{1}{2\eta^2} \|E^{t+1}\|^2 + \frac{\eta^2}{2} G_W^2 \tag{15}$$

We have:

$$f(W^{t+1}) \leq f(W_{\text{ideal}}^{t+1}) + \left( \frac{L}{2} + \frac{1}{2\eta^2} \right) \|E^{t+1}\|^2 + \frac{\eta^2}{2} G_W^2 \tag{16}$$

*Step 2b: Gap between $W_{\text{ideal}}^{t+1}$ and $W^t$*
Applying smoothness between $W_{\text{ideal}}^{t+1}$ and $W^t$:

$$f(W_{\text{ideal}}^{t+1}) \leq f(W^t) + \langle \nabla f(W^t), W_{\text{ideal}}^{t+1} - W^t \rangle + \frac{L}{2} \|W_{\text{ideal}}^{t+1} - W^t\|^2 \tag{17}$$

$$\leq f(W^t) + \langle \nabla f(W^t), W_{\text{ideal}}^{t+1} - W^{t+1} + W^{t+1} - W^t \rangle + \frac{L}{2} \|W_{\text{ideal}}^{t+1} - W^{t+1} + W^{t+1} - W^t\|^2 \tag{18}$$

$$= f(W^t) - \langle \nabla f(W^t), E^{t+1} \rangle + \langle \nabla f(W^t), W^{t+1} - W^t \rangle + L(\|E^{t+1}\|^2 + \|W^{t+1} - W^t\|^2) \tag{19}$$

Following similar steps from (15): $-\langle \nabla f(W^t), E^{t+1} \rangle \leq \frac{1}{2\eta^2} \|E^{t+1}\|^2 + \frac{\eta^2}{2} G_W^2$. Then we have:

$$f(W_{\text{ideal}}^{t+1}) \leq f(W^t) - \langle \nabla f(W^t), E^{t+1} \rangle + \langle \nabla f(W^t), W^{t+1} - W^t \rangle + L(\|E^{t+1}\|^2 + \|W^{t+1} - W^t\|^2) \tag{20}$$

$$\leq f(W^t) + \left( \frac{1}{2\eta^2} \|E^{t+1}\|^2 + \frac{\eta^2}{2} G_W^2 \right) + \langle \nabla f(W^t), W^{t+1} - W^t \rangle + L(\|E^{t+1}\|^2 + \|W^{t+1} - W^t\|^2) \tag{21}$$

*Step 2c: Gap between $W^{t+1}$ and $W^t$*
Substituting (21) into (16):

$$f(W^{t+1}) \leq f(W_{\text{ideal}}^{t+1}) + \left( \frac{L}{2} + \frac{1}{2\eta^2} \right) \|E^{t+1}\|^2 + \frac{\eta^2}{2} G_W^2 \tag{22}$$

$$\leq f(W^t) + (\frac{3L}{2} + \frac{1}{\eta^2}) \|E^{t+1}\|^2 + \langle \nabla f(W^t), W^{t+1} - W^t \rangle + L\|W^{t+1} - W^t\|^2 + \eta^2 G_W^2 \tag{23}$$

The change in global weights is:

$$W^{t+1} - W^t = \bar{B}^{t+1} \bar{A}^{t+1} - \bar{B}^t \bar{A}^t$$

$$= \left( \bar{B}^t - \eta \frac{1}{N} \sum_{i=1}^N \nabla_B f_i(W^t, \xi_i) \right) \left( \bar{A}^t - \eta \frac{1}{N} \sum_{i=1}^N \nabla_A f_i(W^t, \xi_i) \right) - \bar{B}^t \bar{A}^t$$

$$\text{Define } G_A^t = \frac{1}{N} \sum_{i=1}^N \nabla_A f_i(W^t, \xi_i) \text{ and } G_B^t = \frac{1}{N} \sum_{i=1}^N \nabla_B f_i(W^t, \xi_i), \tag{24}$$

$$= (\bar{B}^t \bar{A}^t - \eta \bar{B}^t G_A^t - \eta G_B^t \bar{A}^t + \eta^2 G_B^t G_A^t) - \bar{B}^t \bar{A}^t$$

$$= \underbrace{-\eta(\bar{B}^t G_A^t + G_B^t \bar{A}^t)}_{\text{First Order Term}} + \underbrace{\eta^2 G_B^t G_A^t}_{\text{Second Order Term}} \tag{25}$$

Substituting this expansion into the inner product term $\langle W^{t+1} - W^t, \nabla f(W^t) \rangle$:

$$\langle W^{t+1} - W^t, \nabla f(W^t) \rangle = \langle -\eta(\bar{B}^t G_A^t + G_B^t \bar{A}^t) + \eta^2 G_B^t G_A^t, \nabla f(W^t) \rangle$$

$$= -\eta \underbrace{\langle \bar{B}^t G_A^t, \nabla f(W^t) \rangle}_{\text{Term I}} - \eta \underbrace{\langle G_B^t \bar{A}^t, \nabla f(W^t) \rangle}_{\text{Term II}} + \eta^2 \langle G_B^t G_A^t, \nabla f(W^t) \rangle \tag{26}$$

We apply the chain rule properties of matrix calculus to simplify Terms I and II. Note that $\nabla_A f(W) = (\bar{B}^t)^T \nabla_W f(W)$ and $\nabla_B f(W) = \nabla_W f(W)(\bar{A}^t)^T$.

- **Term I:** Using the trace property $\langle X, Y \rangle = \text{Tr}(X^T Y)$:

$$\langle \bar{B}^t G_A^t, \nabla f(W^t) \rangle = \langle G_A^t, (\bar{B}^t)^T \nabla f(W^t) \rangle = \langle G_A^t, \nabla_A f(W^t) \rangle \tag{27}$$

- **Term II:** Similarly:

$$\langle G_B^t \bar{A}^t, \nabla f(W^t) \rangle = \langle G_B^t, \nabla f(W^t)(\bar{A}^t)^T \rangle = \langle G_B^t, \nabla_B f(W^t) \rangle \tag{28}$$

We bound the higher-order inner product term using the Cauchy-Schwarz inequality and assuming bounded gradients:

$$\langle \eta^2 G_B^t G_A^t, \nabla f(W^t) \rangle \leq \eta^2 \|G_B^t G_A^t\| \|\nabla f(W^t)\| \tag{29}$$

$$\leq \eta^2 G_W \|G_B^t G_A^t\| \tag{30}$$

$$\leq \eta^2 G_W \|G_B^t\| \|G_A^t\| \tag{31}$$

$$\leq \eta^2 G_W G_B G_A \tag{32}$$

We bound $\|W^{t+1} - W^t\|^2$:

$$
\begin{aligned}
\|W^{t+1} - W^t\|^2 &= \| - \eta(\bar{B}^t G_A^t + G_B^t \bar{A}^t) + \eta^2 G_B^t G_A^t\|^2 \\
&\leq 2\eta^2 \|\bar{B}^t G_A^t + G_B^t \bar{A}^t\|^2 + 2\eta^4 \|G_B^t G_A^t\|^2 \\
&\leq \eta^2 M_1 + \eta^4 M_2 \quad \text{(where } M_1, M_2 \text{ are constant combination of bounds)}
\end{aligned}
\tag{33}
$$

Substituting these back into (23):

$$
\begin{aligned}
f(W^{t+1}) \leq f(W^t) &+ (\frac{3L}{2} + \frac{1}{\eta^2})\|E^{t+1}\|^2 \\
&- \eta\langle G_A^t, \nabla_A f(W^t)\rangle - \eta\langle G_B^t, \nabla_B f(W^t)\rangle \\
&+ \eta^2 G_W^2 + \eta^2 (G_W G_A G_B + M_1) + \eta^4 M_2
\end{aligned}
\tag{34}
$$

Taking the expectation $\mathbb{E}$ over the sampling randomness at round $t$, where $\mathbb{E}[G_A^t] = \nabla_A f(W^t)$ and $\mathbb{E}[G_B^t] = \nabla_B f(W^t)$, yields the final recursive step:

$$
\mathbb{E}[f(W^{t+1})] \leq \mathbb{E}[f(W^t)] - \eta\mathbb{E}\left[\|\nabla_A f(W^t)\|^2 + \|\nabla_B f(W^t)\|^2\right] + (\frac{3L}{2} + \frac{1}{\eta^2})\mathbb{E}[\|E^{t+1}\|^2] + O(\eta^2)
\tag{35}
$$

We rearrange the inequality to isolate the gradient norm term on the left-hand side:

$$
\eta\mathbb{E}\left[\|\nabla_A f(W^t)\|^2 + \|\nabla_B f(W^t)\|^2\right] \leq \mathbb{E}[f(W^t)] - \mathbb{E}[f(W^{t+1})] + (\frac{3L}{2} + \frac{1}{\eta^2})\mathbb{E}[\|E^{t+1}\|^2] + O(\eta^2)
\tag{36}
$$

We sum this inequality over the training rounds from $t = 0$ to $T - 1$ (or equivalently 1 to $T$, depending on indexing notation; we use 0 to $T - 1$ to match the standard updates):

$$
\begin{aligned}
\eta\sum_{t=0}^{T-1} \mathbb{E}\left[\|\nabla_A f(W^t)\|^2 + \|\nabla_B f(W^t)\|^2\right] &\leq \sum_{t=0}^{T-1}\left(\mathbb{E}[f(W^t)] - \mathbb{E}[f(W^{t+1})]\right) \\
&+ (\frac{3L}{2} + \frac{1}{\eta^2})\sum_{t=0}^{T-1}\mathbb{E}[\|E^{t+1}\|^2] + \sum_{t=0}^{T-1} O(\eta^2)
\end{aligned}
\tag{37}
$$

The first term on the RHS is a telescoping sum:

$$
\sum_{t=0}^{T-1}\left(\mathbb{E}[f(W^t)] - \mathbb{E}[f(W^{t+1})]\right) = f(W^0) - \mathbb{E}[f(W^T)]
\tag{38}
$$

Let $f^* = \inf_W f(W)$ denote the infimum of the objective. Since $f^* \leq f(W^T)$, we have $f(W^0) - \mathbb{E}[f(W^T)] \leq f(W^0) - f^*$. Substituting this back:

$$
\eta\sum_{t=0}^{T-1} \mathbb{E}\left[\|\nabla_A f(W^t)\|^2 + \|\nabla_B f(W^t)\|^2\right] \leq f(W^0) - f^* + (\frac{3L}{2} + \frac{1}{\eta^2})\sum_{t=0}^{T-1}\mathbb{E}[\|E^{t+1}\|^2] + T \cdot O(\eta^2)
\tag{39}
$$

## 3. Final Bound

We divide the entire inequality by $T\eta$ to obtain the average gradient norm:

$$
\begin{aligned}
\frac{1}{T}\sum_{t=0}^{T-1} \mathbb{E}\left[\|\nabla_A f(W^t)\|^2 + \|\nabla_B f(W^t)\|^2\right] &\leq \frac{f(W^0) - f^*}{T\eta} + (\frac{3L}{2T\eta} + \frac{1}{T\eta^3})\sum_{t=0}^{T-1}\mathbb{E}[\|E^{t+1}\|^2] + \frac{T \cdot O(\eta)}{T\eta} \\
&= \frac{f(W^0) - f^*}{T\eta} + \frac{3L\eta^2 + 2}{2T\eta}\sum_{t=0}^{T-1}\mathbb{E}\left[\frac{\|E^{t+1}\|^2}{\eta^2}\right] + O(\eta)
\end{aligned}
\tag{40}
$$

Using the property that the minimum of a set is less than or equal to its average, i.e.,

$$\min_{t \in \{0,\ldots,T-1\}} \mathbb{E}\left[\|\nabla_A f(W^t)\|^2 + \|\nabla_B f(W^t)\|^2\right] \leq \frac{1}{T}\sum_{t=0}^{T-1} \mathbb{E}\left[\|\nabla_A f(W^t)\|^2 + \|\nabla_B f(W^t)\|^2\right] \tag{41}$$

We show the convergence:

$$\min_t \mathbb{E}\left[\|\nabla_A f(W^t)\|^2 + \|\nabla_B f(W^t)\|^2\right] \leq \frac{f(W^0) - f^*}{T\eta} + \frac{3L\eta^2 + 2}{2T\eta}\sum_{t=0}^{T-1} \mathbb{E}\left[\frac{\|E^{t+1}\|^2}{\eta^2}\right] + O(\eta) \tag{42}$$

$\square$

## A.2. Proof of Theorem 4.8 and Corollary 4.9

**Lemma A.1** (Soft Rotation Shrinkage). *Let $R$ be a rotation matrix and $\lambda \in [0, 1]$. Define the soft rotation matrix $R_{soft}$ as in Eq. (5). Then*

$$\|R_{\text{soft}} - I\|_F \leq 2\lambda\|R - I\|_F. \tag{43}$$

*Consequently, in A-alignment rounds, with FedRot-LoRA, we have*

$$\|R_{i,\text{soft}}^t - I\|_F \leq 2\lambda\|R_i^{t,*} - I\|_F \leq 2\kappa\lambda\|A_i^t - A_{\text{ref}}\|_F \tag{44}$$

*where $\kappa_\lambda = 2\kappa\lambda$. Analogous results hold for B-alignment rounds.*

*Proof.* Notice that Eq. (5) is the closed-form solution for the problem:

$$R_{\text{soft}} = \arg\min_Q \left\|Q - \left((1-\lambda)I + \lambda R\right)\right\|_F^2, \tag{45}$$

$$\text{s.t. } Q^\top Q = I, \det(Q) > 0. \tag{46}$$

Define

$$M(\lambda) = (1 - \lambda)I + \lambda R. \tag{47}$$

By definition, $R_{\text{soft}}$ minimizes $\|Q - M(\lambda)\|_F$ with constraints (46). Notice that $I$ is a feasible solution, then

$$\|R_{\text{soft}} - M(\lambda)\|_F \leq \|I - M(\lambda)\|_F. \tag{48}$$

But $I - M(\lambda) = I - [(1-\lambda)I + \lambda R] = \lambda(I - R)$, hence $\|I - M(\lambda)\|_F = \lambda\|R - I\|_F$. Using triangle inequality,

$$\|R_{\text{soft}} - I\|_F \leq \|R_{\text{soft}} - M(\lambda)\|_F + \|M(\lambda) - I\|_F \leq \lambda\|R - I\|_F + \lambda\|R - I\|_F = 2\lambda\|R - I\|_F, \tag{49}$$

proving (43). With Assumption 4.7, the final inequality holds:

$$\|R_{i,\text{soft}}^t - I\|_F \leq 2\lambda\|R_i^{t,*} - I\|_F \tag{50}$$

$$\leq 2\lambda\kappa\|A_i^t - A_{\text{ref}}\|_F \tag{51}$$

$\square$

**Proof of Theorem 4.8**

*Proof.* The error gap is defined as:

$$E^t = W^t - W_{\text{ideal}}^t \tag{52}$$

$$= \left(\frac{1}{N}\sum_{i=1}^N B_i^t\right)\left(\frac{1}{N}\sum_{i=1}^N A_i^t\right) - \frac{1}{N}\sum_{i=1}^N B_i^t A_i^t \tag{53}$$

$$= -\frac{1}{2N^2}\sum_{i=1}^N\sum_{j=1}^N (B_i^t - B_j^t)(A_i^t - A_j^t) \quad \text{(Apply Lagrange's Identity)} \tag{54}$$

Thus, the norm of $E^t$ is:

$$\|E^t\| = \left\| \frac{1}{2N^2} \sum_{i=1}^{N} \sum_{j=1}^{N} (B_i^t - B_j^t)(A_i^t - A_j^t) \right\| \tag{55}$$

$$\leq \frac{1}{2N^2} \sum_{i=1}^{N} \sum_{j=1}^{N} \left\| (B_i^t - B_j^t)(A_i^t - A_j^t) \right\| \qquad \text{(Triangle Inequality)} \tag{56}$$

$$\leq \frac{1}{2N^2} \sum_{i=1}^{N} \sum_{j=1}^{N} \left\| B_i^t - B_j^t \right\| \cdot \left\| A_i^t - A_j^t \right\| \tag{57}$$

Let $x_{ij} := \|B_i^t - B_j^t\|$ and $y_{ij} := \|A_i^t - A_j^t\|$. Cauchy–Schwarz gives $\left(\sum_{i,j} x_{ij} y_{ij}\right)^2 \leq \left(\sum_{i,j} x_{ij}^2\right)\left(\sum_{i,j} y_{ij}^2\right)$. Thus

$$\|E^t\|^2 \leq \left( \frac{1}{2N^2} \sum_{i,j} \|B_i^t - B_j^t\|^2 \right) \left( \frac{1}{2N^2} \sum_{i,j} \|A_i^t - A_j^t\|^2 \right) \tag{58}$$

Apply AM-GM inequality ($\sqrt{xy} \leq \frac{1}{2}(x+y)$),

$$\|E^t\| \leq \frac{1}{2} \left( \frac{1}{2N^2} \sum_{i,j} \|B_i^t - B_j^t\|^2 + \frac{1}{2N^2} \sum_{i,j} \|A_i^t - A_j^t\|^2 \right) \tag{59}$$

For each pair $(i, j)$,

$$\|A_i^t - A_j^t\|^2 = \|(A_i^t - A_{\text{ref}}) - (A_j^t - A_{\text{ref}})\|^2 \leq 2\|A_i^t - A_{\text{ref}}\|^2 + 2\|A_j^t - A_{\text{ref}}\|^2, \tag{60}$$

Sum over $i, j$ and divide by $2N^2$:

$$\frac{1}{2N^2} \sum_{i,j} \|A_i^t - A_j^t\|^2 \leq \frac{1}{2N^2} \sum_{i,j} \left( 2\|A_i^t - A_{\text{ref}}\|^2 + 2\|A_j^t - A_{\text{ref}}\|^2 \right). \tag{61}$$

Since $\sum_{j=1}^{N} 1 = N$,

$$\frac{1}{2N^2} \sum_{i,j} 2\|A_i^t - A_{\text{ref}}\|^2 = \frac{1}{N} \sum_{i=1}^{N} \|A_i^t - A_{\text{ref}}\|^2, \tag{62}$$

and similarly for the $j$-term. Hence

$$\frac{1}{2N^2} \sum_{i,j} \|A_i^t - A_j^t\|^2 \leq \frac{2}{N} \sum_{i=1}^{N} \|A_i^t - A_{\text{ref}}\|^2. \tag{63}$$

Similar inequalities hold for matrix $B$:

$$\frac{1}{2N^2} \sum_{i,j} \|B_i^t - B_j^t\|^2 \leq \frac{2}{N} \sum_{i=1}^{N} \|B_i^t - B_{\text{ref}}\|^2. \tag{64}$$

Apply (63) and (64) to (59),

$$\|E^t\| \leq \frac{1}{2} \left( \frac{1}{2N^2} \sum_{i,j} \|B_i^t - B_j^t\|^2 + \frac{1}{2N^2} \sum_{i,j} \|A_i^t - A_j^t\|^2 \right) \tag{65}$$

$$\leq \frac{1}{N} \sum_{i=1}^{N} \|B_i^t - B_{\text{ref}}\|^2 + \frac{1}{N} \sum_{i=1}^{N} \|A_i^t - A_{\text{ref}}\|^2 \tag{66}$$

With Assumption 4.5, we have:

$$\frac{1}{N} \sum_{i=1}^{N} \|\tilde{A}_i^t - A_{\text{ref}}\|^2 \leq \frac{1}{N} (1 - \alpha(\lambda)) \sum_{i=1}^{N} \|A_i^t - A_{\text{ref}}\|^2 \tag{67}$$

Here we bound $\|\tilde{B}_i^t - B_{\text{ref}}\|^2$.

$$\|\tilde{B}_i^t - B_{\text{ref}}\|^2 = \|\tilde{B}_i^t - B_i^t\|^2 + \|B_i^t - B_{\text{ref}}\|^2 + 2\langle \tilde{B}_i^t - B_i^t, B_i^t - B_{\text{ref}} \rangle \tag{68}$$

With Lemma A.1: $\|R_{i,\text{soft}}^t - I\| \leq 2\lambda \|R_i^{t,*} - I\| \leq 2\kappa\lambda \|A_i^t - A_{\text{ref}}\|$, then:

$$\|\tilde{B}_i^t - B_i^t\|^2 \leq \|B_i^t\|^2 \|R_{i,\text{soft}}^t - I\|^2 \tag{69}$$

$$\leq \|B_i^t\|^2 (2\kappa\lambda)^2 \|A_i^t - A_{\text{ref}}\|^2 \tag{70}$$

Here we provide bounds for $\|B_i^t\|$ and $\|A_i^t\|$.

*Remark* A.2 (Rescaling Does Not Affect Rotational Alignment). Due to the scale invariance of LoRA, the factors $A_i^t$ and $B_i^t$ can be rescaled without changing the effective update $\Delta W_i^t = B_i^t A_i^t$. In particular, for any $c > 0$, one may apply the transformation

$$B_i^{t'} = \tfrac{1}{c} B_i^t, \qquad A_i^{t'} = c A_i^t, \qquad B_{\text{ref}}' = \tfrac{1}{c} B_{\text{ref}}, \qquad A_{\text{ref}}' = c A_{\text{ref}}. \tag{71}$$

This rescaling leaves the Procrustes alignment problem unchanged, since both the local factors and the reference are scaled consistently, and therefore does not affect the resulting rotation or the aggregated update. Such rescaling can be reversed after aggregation and thus has no impact on the final model update. We introduce this operation solely for theoretical analysis to rule out degenerate factorizations with extreme magnitudes (e.g., $B \to 0$, $A \to \infty$). In practice, we do not perform explicit rescaling, as the norms of $A$ and $B$ remain naturally bounded due to the limited number of local fine-tuning steps.

With Remark A.2 and Assumption 4.3 ($\|\Delta W_i^t\| \leq \tau$), we choose $c = \sqrt{\frac{\|A_i^t\|}{\|B_i^t\|}}$ for rescaling, leading to $\|cB_i^t\| = \|\frac{1}{c}A_i^t\|$. The rescaled factors are bounded:

$$\|A_i^t\| = \|B_i^t\| \leq \sqrt{\tau}. \tag{72}$$

We further bound Eq.(70):

$$\|\tilde{B}_i^t - B_i^t\|^2 \leq \tau (2\kappa\lambda)^2 \|A_i^t - A_{\text{ref}}\|^2 \tag{73}$$

Then we bound the term $2\langle \tilde{B}_i^t - B_i^t, B_i^t - B_{\text{ref}} \rangle$ from Eq. (68).

$$2\langle \tilde{B}_i^t - B_i^t, B_i^t - B_{\text{ref}} \rangle \leq 2 \cdot \|\tilde{B}_i^t - B_i^t\| \cdot \|B_i^t - B_{\text{ref}}\| \tag{74}$$

$$\leq 2 \cdot \left(2\sqrt{\tau}\kappa\lambda\right) \|A_i^t - A_{\text{ref}}\| \cdot \|B_i^t - B_{\text{ref}}\| \tag{75}$$

$$= 4\sqrt{\tau}\kappa\lambda \|A_i^t - A_{\text{ref}}\| \cdot \|\eta \nabla_B f_i(W^{t-1})\| \tag{76}$$

$$\leq 4\sqrt{\tau}\kappa\lambda\eta G_B \|A_i^t - A_{\text{ref}}\| \tag{77}$$

Thus, Eq. (68) can be bounded as:

$$\|\tilde{B}_i^t - B_{\text{ref}}\|^2 = \|\tilde{B}_i^t - B_i^t\|^2 + \|B_i^t - B_{\text{ref}}\|^2 - 2\langle (R_{i,\text{soft}}^t - I)A_i^t, \eta G_{A,i}^t \rangle \tag{78}$$

$$\leq \|B_i^t - B_{\text{ref}}\|^2 + 4\tau\kappa^2\lambda^2 \|A_i^t - A_{\text{ref}}\|^2 + 4\sqrt{\tau}\kappa\lambda\eta G_B \underbrace{\|A_i^t - A_{\text{ref}}\|}_{\|A_i^t - A_{\text{ref}}\| \geq \delta_A} \tag{79}$$

$$\leq \|B_i^t - B_{\text{ref}}\|^2 + \left(4\tau\kappa^2\lambda^2 + \frac{4}{\delta_A}\sqrt{\tau}\kappa\lambda\eta G_B\right) \|A_i^t - A_{\text{ref}}\|^2 \tag{80}$$

Apply (67) and (80) to (66),

$$\|E^t\| \leq \frac{1}{N} \sum_{i=1}^{N} \|\tilde{B}_i^t - B_{\text{ref}}\|^2 + \frac{1}{N} \sum_{i=1}^{N} \|\tilde{A}_i^t - A_{\text{ref}}\|^2 \quad \text{(Error bound with rotation alignment)} \tag{81}$$

$$\leq \frac{1}{N} \sum_{i=1}^{N} \left( \underbrace{\|B_i^t - B_{\text{ref}}^2\|^2 + \|A_i^t - A_{\text{ref}}\|^2}_{\text{error bound without rotation}} - \left( \alpha(\lambda) - \left( 4\tau\kappa^2\lambda^2 + \frac{4}{\delta_A}\sqrt{\tau}\kappa\lambda\eta G_B \right) \right) \|A_i^t - A_{\text{ref}}\|^2 \right) \tag{82}$$

Notice that with Assumption 4.5, we have $\alpha(\lambda) > c_0 \lambda$, therefore,

$$\|E^t\| \leq \frac{1}{N} \sum_{i=1}^{N} \left( \underbrace{\|B_i^t - B_{\text{ref}}^2\|^2 + \|A_i^t - A_{\text{ref}}\|^2}_{\text{error bound without rotation}} - \left( c_0\lambda - \left( 4\tau\kappa^2\lambda^2 + \frac{4}{\delta_A}\sqrt{\tau}\kappa\lambda\eta G_B \right) \right) \|A_i^t - A_{\text{ref}}\|^2 \right) \tag{83}$$

To ensure positive gain, we require $c_0\lambda - \left( 4\tau\kappa^2\lambda^2 + \frac{4}{\delta_A}\sqrt{\tau}\kappa\lambda\eta G_B \right) > 0$. Rearranging this term yields the quadratic inequality

$$4\kappa^2\tau\lambda^2 + \left( \frac{4}{\delta_A}\sqrt{\tau}\kappa\eta G_B - c_0 \right)\lambda < 0. \tag{84}$$

Since $\lambda > 0$, a necessary condition for (84) to admit a feasible solution is $\frac{4}{\delta_A}\sqrt{\tau}\kappa\eta G_B - c_0 < 0$, which equivalently requires

$$\eta < \frac{c_0\delta_A}{4\sqrt{\tau}\kappa G_B}. \tag{85}$$

Under (85), the set of $\lambda$ values that ensure a positive gain is given by

$$0 < \lambda < \min\left\{ 1, \frac{c_0\delta_A - 4\sqrt{\tau}\kappa\eta G_B}{4\kappa^2\tau\delta_A} \right\}. \tag{86}$$

This completes the proof. $\square$

# B. Additional Experimental Details

## B.1. Comparison of federated LoRA methods

*Table 7.* Comparison of representative federated LoRA methods in terms of aggregation type, aggregation space, communication payload, and additional computation. "Pure agg." indicates whether the method mainly modifies the aggregation rule without introducing personalization, compression/reconstruction, or residual-transmission components. Computational costs are reported per communication round unless otherwise specified.

| Method | Pure agg. | Agg. space | Communication | Extra computation |
|---|---|---|---|---|
| FlexLoRA | Yes | Full-parameter space | LoRA factors $B, A$ only | Full parameter SVD: $O(d^3)$ |
| FedSRD | No | Full-parameter space | Sparsified factors $O(\rho dr), \rho < 1$ | Sparsification + SVD: $O(d^3)$ |
| FedEx-LoRA | Yes | Mixed $\bar{B}\bar{A} + \Delta W_{res}$ | Residual matrix $\Delta W_{res} \in \mathbb{R}^{d \times d}$ | $O(Nd^2 r)$ to form residual matrix |
| LoRA-FAIR | Yes | Full-parameter space | LoRA factors $B, A$ only | $O(Td^2 r)$ for $T$ steps gradient descent |
| FLoRA | Yes | Full-parameter space | Stacked $B \in \mathbb{R}^{d \times Nr}, A \in \mathbb{R}^{Nr \times d}$ | server stacking $O(Nrd)$ |
| FedSA-LoRA | No | Factor $A$ only (personalized FL) | Factor $A$ only | 0 |

## B.2. Detailed Experimental Settings

Table 8 summarizes the hyperparameters used in our experiments. All results are averaged over 3 random seeds. We perform a grid search over the learning rate $\eta \in \{5e\text{-}4, 1e\text{-}3, 5e\text{-}3, 2e\text{-}2\}$. For FedRot-LoRA, we tune the alignment strength $\lambda \in \{0.2, 0.4, 0.6, 0.8, 1.0\}$. We set the number of local epochs to 20 and the number of communication rounds to 250 for all natural language understanding tasks, and to 30 local epochs and 200 communication rounds for generative tasks. For each method, hyperparameters are tuned separately using validation, and the best-performing configurations are reported in Table 8. The aggregation error reported in Table 2 is computed from the MNLI results in Table 1 with $N = 3$ clients. We sum the aggregation error across all LoRA layers and average over all communication rounds and 3 random seeds. For Figure 3, we additionally evaluate $\lambda \in \{0.1, 0.3, 0.5, 0.7, 0.9\}$. The results in Table 6 use the same experimental configurations as Table 1 ($N = 3$, MNLI). For MNLI experiments, all results in this paper correspond to the matched (MNLI-m) setting. Most of our experiments are conducted under highly non-IID data distributions. We perform HumanEval (trained on CodeSearchNet) and GSM8K experiments (Table 5) following the default configurations of (Kuang et al., 2024). Since LoRA is initialized with $\bar{B}^0 \bar{A}^0 = \mathbf{0}$, FedRot-LoRA applies rotational alignment after the first communication round to avoid ill-conditioned rotations.

*Table 8.* The hyperparameters for each experimental setting.

| Experiments | Clients Num | Rank | Data distributions | Optimal learning rates (FedIT/FFA-LoRA/RoLRA/FedRot-LoRA) | | Optimal $\lambda$ |
|---|---|---|---|---|---|---|
| GLUE results (Table 1) | $N = 3, 10$ | $r = 4$ | Dirichlet $h = 0.5$ | SST-2 | $N = 3 : (0.02/0.02/0.0005/0.02)$ | $\lambda = 0.6$ |
| | | | | | $N = 10 : (0.005/0.02/0.0005/0.005)$ | $\lambda = 0.6$ |
| | | | | QNLI | $N = 3 : (0.02/0.02/0.001/0.005)$ | $\lambda = 0.2$ |
| | | | | | $N = 10 : (0.02/0.02/0.0005/0.005)$ | $\lambda = 0.2$ |
| | | | | QQP | $N = 3 : (0.005/0.02/0.001/0.005)$ | $\lambda = 0.4$ |
| | | | | | $N = 10 : (0.005/0.02/0.0005/0.005)$ | $\lambda = 0.6$ |
| | | | | RTE | $N = 3 : (0.005/0.02/0.0005/0.005)$ | $\lambda = 0.2$ |
| | | | | | $N = 10 : (0.005/0.02/0.0005/0.005)$ | $\lambda = 0.4$ |
| | | | | MNLI | $N = 3 : (0.02/0.02/0.001/0.005)$ | $\lambda = 0.4$ |
| | | | | | $N = 10 : (0.005/0.02/0.001/0.005)$ | $\lambda = 0.4$ |
| MNLI with different ranks (Table 3) | $N = 3$ | $r = 4, 8, 16$ | Dirichlet $h = 0.5$ | $r = 4 : (0.02/0.02/0.001/0.005)$ | | $\lambda = 0.4$ |
| | | | | $r = 8 : (0.02/0.02/0.001/0.005)$ | | $\lambda = 0.4$ |
| | | | | $r = 16 : (0.02/0.02/0.0005/0.005)$ | | $\lambda = 0.4$ |
| MNLI with different non-IID level (Table 4) | $N = 3$ | $r = 4$ | Dirichlet $h = 100, 1, 0.5$ | $h = 100 : (0.005/0.02/0.001/0.005)$ | | $\lambda = 0.8$ |
| | | | | $h = 1 : (0.02/0.02/0.001/0.005)$ | | $\lambda = 0.8$ |
| | | | | $h = 0.5 : (0.02/0.02/0.001/0.005)$ | | $\lambda = 0.4$ |
| HumanEval (Trained on CodeSearchNet, Table 5) | $N = 6$ | $r = 8$ | Non-IID: distinct programming languages per client | $(0.005/0.005/0.0005/0.005)$ | | $\lambda = 0.4$ |
| GSM8K (Table 5) | $N = 3$ | $r = 8$ | IID | $(0.005/0.005/0.001/0.005)$ | | $\lambda = 0.2$ |

*Table 9.* Comparison of scalar rescaling and rotational alignment. Results show that rotational alignment substantially outperforms rescaling in higher-dimensional LoRA settings.

| Methods | SST-2 | MNLI | GSM8K |
|---|---|---|---|
| FedIT (No Alignment) | $0.953 \pm 0.001$ | $0.866 \pm 0.001$ | $0.4293 \pm 0.018$ |
| Scalar Rescaling | $0.953 \pm 0.002$ | $0.865 \pm 0.006$ | $0.4286 \pm 0.021$ |
| FedRot-LoRA (Rotation) | $0.954 \pm 0.001$ | $0.876 \pm 0.002$ | $0.4437 \pm 0.009$ |

## B.3. Additional Experimental Results

### B.3.1. RESCALING VS ROTATION

In Section 3, we introduced a scalar optimization example (Figure 2) to illustrate the effect of aggregation misalignment in federated LoRA. This example is one-dimensional and serves to show that naive factor-wise aggregation can introduce instability near the optimal solution manifold. In this example, we consider three clients with local optima satisfying $B_1 A_1 = 0.5$, $B_2 A_2 = 1.0$, and $B_3 A_3 = 1.5$, so that the global optimum is $BA = 1.0$. We set the learning rate as $0.01$, and local steps as 30 per communication round. In this setting, FedIT (naive factor-wise averaging) suffers from aggregation misalignment: when the global iterate approaches the optimal manifold, aggregation noise causes the parameters to drift along the manifold, leading to misguided slide along the curve and slow convergence. Factor-freezing methods avoid this issue by enforcing linear aggregation, but at the cost of reduced expressivity and slower optimization.

In one-dimension, the only admissible invariance of the factorization is *scalar rescaling*. To mitigate aggregation misalignment while allowing both factors to be trained, we consider aligning local updates via a scalar transformation. Specifically, in round $t$, each client solves

$$\min_{c_i^t} \|c_i^t A_i^t - A_{\text{ref}}\|^2, \quad \text{if } t \bmod 2 = 1 \tag{87}$$

$$\min_{c_i^t} \|c_i^t B_i^t - B_{\text{ref}}\|^2, \quad \text{if } t \bmod 2 = 0 \tag{88}$$

where $(A_{\text{ref}}, B_{\text{ref}})$ denote the aggregated global parameters from the previous round. The above problem admits closed-form solutions,

$$c_i^t = \frac{\langle A_i^t, A_{\text{ref}} \rangle}{\|A_i^t\|_F^2}, \quad \text{if } t \bmod 2 = 1 \tag{89}$$

$$c_i^t = \frac{\langle B_i^t, B_{\text{ref}} \rangle}{\|B_i^t\|_F^2}, \quad \text{if } t \bmod 2 = 0 \tag{90}$$

The aligned factors are given by

$$\tilde{A}_i^t = c_i^t A_i^t, \quad \tilde{B}_i^t = (c_i^t)^{-1} B_i^t, \quad \text{if } t \bmod 2 = 1 \tag{91}$$

$$\tilde{A}_i^t = (c_i^t)^{-1} A_i^t, \tilde{B}_i^t = c_i^t B_i^t. \quad \text{if } t \bmod 2 = 0 \tag{92}$$

In the scalar case, this rescaling exactly preserves the semantic update ($\tilde{B}_i^t \tilde{A}_i^t = B_i^t A_i^t$) and is sufficient to eliminate aggregation misalignment.

In higher-dimensional LoRA, $\Delta W = BA$ with $B \in \mathbb{R}^{d \times r}$ and $A \in \mathbb{R}^{r \times d}$ for $r > 1$. A natural generalization of scalar rescaling is to replace $c_i^t \in \mathbb{R}$ with a matrix $R_i^t \in \mathbb{R}^{r \times r}$ and consider transformations of the form $B_i^t R_i^t (R_i^t)^{-1} A_i^t$. To preserve semantic equivalence, $R_i^t$ must be invertible, which introduces two major difficulties. First, enforcing invertibility requires nonconvex constraints (e.g., $\det(R_i^t) \neq 0$), and the resulting optimization no longer admits an efficient closed-form solution. Second, unconstrained invertible matrices can be arbitrarily ill-conditioned, leading to numerical instability and highly imbalanced factors during training. As a result, directly optimizing over general invertible transformations is computationally expensive, ill-conditioned, and unsuitable for federated settings.

An alternative is to retain scalar rescaling in high dimensions, effectively restricting $R_i^t = c_i^t I$. While this remains well-defined, it only adjusts the global magnitude of the factors and cannot resolve *subspace misalignment*: rescaling does not

*Table 10.* Effect of reference model selection.

| Reference Model | SST-2 | MNLI |
|---|---|---|
| Random client local model | $0.954 \pm 0.002$ | $0.871 \pm 0.006$ |
| Old global model ($W^{t-2}$) | $0.951 \pm 0.001$ | $0.866 \pm 0.002$ |
| Previous global model ($W^{t-1}$) | $0.954 \pm 0.001$ | $0.876 \pm 0.002$ |

change the column space of $B_i^t$ or the row space of $A_i^t$. As the LoRA rank increases, client updates typically span misaligned latent subspaces, and scalar rescaling alone provides limited reduction in aggregation error. FedRot-LoRA instead restricts $R_i^t$ to the orthogonal group, enforcing $(R_i^t)^\top R_i^t = I$ and $\det(R_i^t) > 0$. This choice ensures invertibility, preserves the semantic update, and yields a well-conditioned transformation with an efficient closed-form solution via the orthogonal Procrustes problem. Orthogonal alignment provides $r(r-1)/2$ degrees of freedom,[2] enabling direct alignment of latent subspaces rather than mere magnitude adjustment. In contrast, scalar rescaling has only one degree of freedom and cannot correct subspace mismatch.

We compare scalar rescaling and FedRot-LoRA in high-dimensional settings on SST-2, MNLI (the same setting of Table 1, $N = 3$) and GSM8K (the same setting of Table 5). For *scalar rescaling*, we perform the same grid search over learning rates as stated in Appendix B.2. Its optimal learning rates are $(0.02, 0.005, 0.005)$ for SST-2, MNLI, and GSM8K, respectively. As reported in Table 9, FedRot-LoRA achieves higher accuracy and greater stability across runs than *scalar rescaling*, confirming that correcting subspace misalignment via rotation is essential in higher-dimensional LoRA.

### B.3.2. EFFECT OF REFERENCE MODEL SELECTION

In the main paper, we use the global model from the previous communication round as the reference model, as this choice has no additional communication overhead. Here, we evaluate alternative reference selection strategies. First, one may use an older global model as the reference (e.g., $W^{t-2}$), which can improve robustness to communication delays or failures. Second, one may use a client's local model as the reference. For this baseline, we randomly sample a client in each round and treat its post-training LoRA factors as the reference. Table 10 compares these strategies. All three choices yield competitive performance. Using either a random client's local model or the previous global model ($W^{t-1}$) consistently outperforms using an older global model ($W^{t-2}$). The random client local model performs well because its update is recent, and aligning to a fresh subspace can be beneficial. However, randomly switching the reference client across rounds may introduce instability, and identifying an ideal fixed client is non-trivial. Using the previous global model ($W^{t-1}$) incurs no additional communication cost and performs well due to the smooth evolution of the LoRA subspace across adjacent rounds; we therefore adopt this strategy by default.

### B.3.3. COMPARISON WITH RANDOM ROTATION

To isolate whether the gains of FedRot-LoRA come from optimized subspace alignment rather than merely exploiting LoRA's rotational invariance, we include a *Random Rotation* baseline. In this variant, each client independently applies a Haar-uniform random rotation to its local LoRA factors prior to aggregation at every communication round.

For client $i$ at round $t$, we generate a random rotation matrix $R_i^t \in \mathrm{SO}(r)$ as follows. We first sample a matrix $Z \in \mathbb{R}^{r \times r}$ with i.i.d. entries $Z_{a,b} \sim \mathcal{N}(0,1)$ and compute its QR decomposition $Z = QP$, where $Q^\top Q = I$ and $P$ is upper triangular. To remove the sign ambiguity inherent in the QR factorization, we fix the column signs of $Q$ by enforcing a nonnegative diagonal of $P$. If $\det(Q) < 0$, we further flip the sign of one column to ensure $\det(Q) = +1$. The resulting matrix $R_i^t := Q$ is Haar-uniformly distributed over the special orthogonal group $\mathrm{SO}(r)$. $R_i^t$ satisfies $(R_i^t)^\top R_i^t = I$ and $\det(R_i^t) = 1$. $R_i^t$ is then applied to the post-training factors:

$$\tilde{B}_i^t = B_i^t R_i^t, \quad \tilde{A}_i^t = (R_i^t)^\top A_i^t. \tag{93}$$

In the random rotation baseline, each client independently samples a rotation matrix $R_i^t$ at every round using the procedure above. In Table 11, we compare FedRot-LoRA with this baseline on SST-2 and MNLI. Despite preserving each client's local update ($\tilde{B}_i^t \tilde{A}_i^t = B_i^t A_i^t$), random rotation leads to unstable training and often collapses performance, as the rotations

---

[2]An arbitrary $r \times r$ matrix has $r^2$ free parameters. The constraint $R^\top R$ imposes $\frac{r(r+1)}{2}$ independent constraints, therefore the remaining degrees of freedom are $\frac{r(r-1)}{2}$.

*Table 11.* Comparison with random rotation. FedRot-LoRA outperforms random rotation on GLUE tasks.

| Methods | SST-2 | MNLI |
|---|---|---|
| Random Rotation | $0.541 \pm 0.182$ | $0.318 \pm 0.000$ |
| FedRot-LoRA | $0.954 \pm 0.001$ | $0.876 \pm 0.002$ |

may amplify cross-client subspace mismatch during aggregation. Unlike FedRot-LoRA, this baseline uses no reference model and optimizes no Procrustes objective; consequently, it does not actively reduce alignment error across clients. FedRot-LoRA explicitly mitigates aggregation misalignment, resulting in substantially more stable and effective training.

### B.3.4. ADDITIONAL RESULTS ON THE EFFECT OF SOFT ROTATION LEVEL $\lambda$

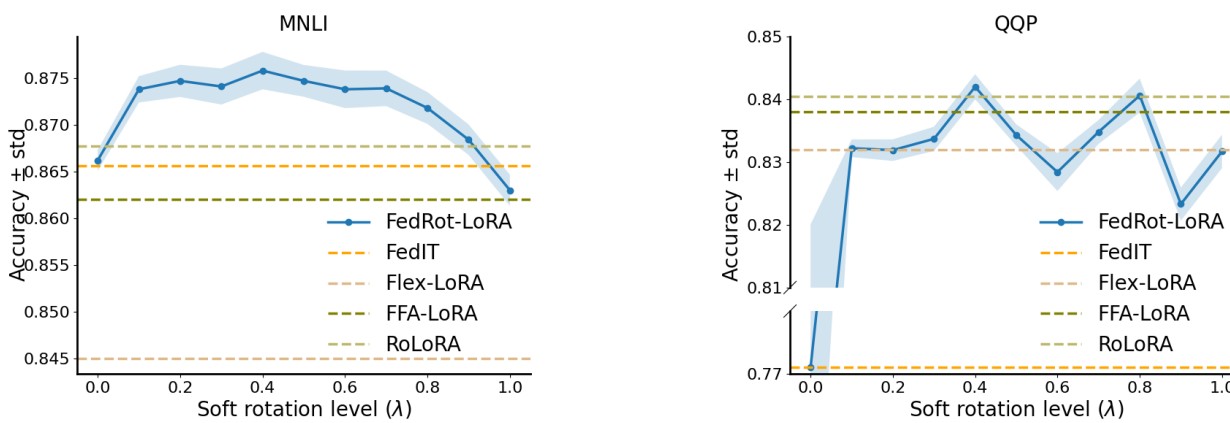

*Figure 4.* Effect of soft rotation level $\lambda$ in FedRot-LoRA on MNLI and QQP.

### B.3.5. EMPIRICAL DIAGNOSTICS FOR THE ALIGNMENT ASSUMPTIONS

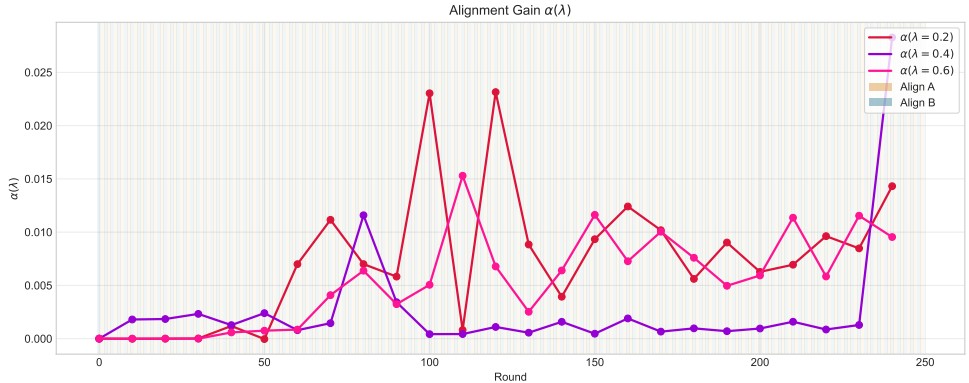

*Figure 5.* Alignment gain $\alpha(\lambda)$ across training rounds. The minimum observed alignment gain is $\alpha(\lambda) \geq 0.00043$. The shaded background regions denote the active alignment target during each respective round. The experimental configuration utilizes the SST-2 dataset distributed across 3 clients, consistent with the settings detailed in Table 1.

Figures 5–7 provide empirical diagnostics for the alignment assumptions used in the analysis. Figure 5 tracks the relative alignment gain $\alpha(\lambda)$ during training and shows that it remains positive for the tested values of $\lambda$, supporting the positive-gain condition in Assumption 4.5. Figure 6 reports the maximum relative rotation discrepancy $\max_i \|R_i^{t,*} - I\|_F / \|A_i^t - A_{\mathrm{ref}}\|_F$, which remains bounded throughout training, consistent with the bounded-dispersion condition in Assumption 4.7. Figure 7 shows that the client factors maintain non-vanishing distance from the global reference, indicating persistent client drift

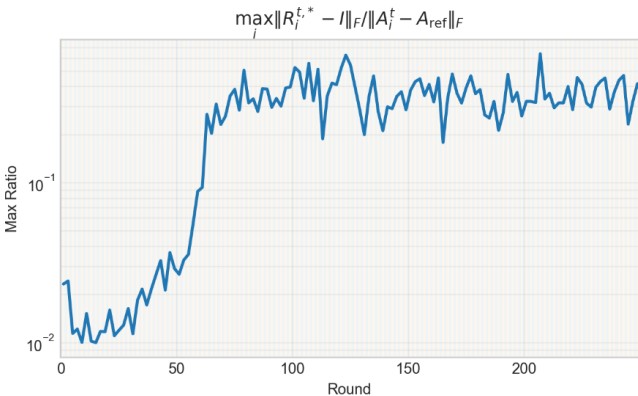

*Figure 6.* Evolution of the maximum relative alignment discrepancy, given by $\max_i \|R_i^{t,*} - I\|_F / \|A_i^t - A_{\text{ref}}\|_F$, among all clients over the course of training rounds $t$. The y-axis is presented on a logarithmic scale to capture the magnitude shifts. The experimental configuration utilizes the SST-2 dataset distributed across 3 clients, consistent with the settings detailed in Table 1.

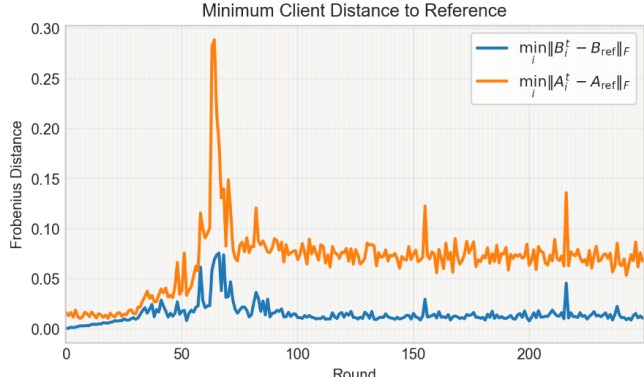

*Figure 7.* Evolution of the minimum client distance to the global reference matrices, quantified by the Frobenius norms $\min_i \|A_i^t - A_{\text{ref}}\|_F$ and $\min_i \|B_i^t - B_{\text{ref}}\|_F$, among all clients at each training round $t$. The experimental configuration utilizes the SST-2 dataset distributed across 3 clients, consistent with the settings detailed in Table 1.

under the non-IID partition and supporting Assumption 4.6. Together, these diagnostics suggest that the assumptions are consistent with observed training behavior in heterogeneous federated settings, rather than relying on pathological or degenerate cases.

