# OpenReview forum: "FedRot-LoRA: Mitigating Rotational Misalignment in Federated LoRA"
_ICML.cc/2026/Conference — ICML 2026 regular_

### Official Review · Reviewer_miTu · 2026-03-12

**Soundness:** 3
**Presentation:** 3
**Significance:** 3
**Originality:** 3
**Overall Recommendation:** 4
**Confidence:** 3

**Summary:**

The paper builds upon prior work on the domain of federated learning, specifically Low-Rank Adaptation (LoRA) finetuning of large language models on decentralized data. The authors claim that current LLM finetuning through federated LoRA techniques carry a non-trivial error during local updates, which accumulates and leads to degraded training performance. This source of this issue, as identified by the authors, is rotational misalignment caused by rotational invariance of low-rank factorizations. To that end, the authors propose FedRot-LoRA, a federated LoRA technique that integrates an alignment step before aggregating the local updates for addressing the identified issue. The submitted work contains a theoretical and experimental analysis to show the effectiveness of the proposed approach against existing established methods.

**Compliance With Llm Reviewing Policy:**

Affirmed.

**Key Questions For Authors:**

1. Performance is stable while rank increases, with some slight degradation when going from rank=8->16. What about lower ranks (e.g. rank=2) or higher (>16)?
2. Could an accuracy/loss convergence curve during training be included, to see how the learning changes with respect to the baselines?
3. In Figure 3, maybe include the individual curves of the baselines instead of their average?

**Limitations:**

Yes

**Strengths And Weaknesses:**

Soundness:

This work is well-articulated with respect to the research problem that it identifies, as well as to the proposed methodology that addresses it. The given theoretical setup is appropriate, along with experimental results/convergence analysis using relevant models and datasets. The presented results and insights gained from comparing with appropriate baselines are credible. That said, it would be nice to see an accuracy/loss curve on the natural language understanding and generation tasks between FedRot-LoRA and the baselines, to understand how the learning/training process itself is affected (e.g. are there spikes during training, does it improve more in the earlier or later stages, etc.).
Additionally, it would also be interesting to see how the complexity that is explained in “Complexity Analysis” translates into actual computational overhead, training time, and convergence time, during training for the various LoRA ranks. This way, for example, one could benefit from knowing the tradeoff between compute-convergence and be able to decide which configuration is more suitable for their use-cases (e.g. under ultra resource-constrained environments).

Presentation:

The presentation of the work is solid, following a straightforward structure that makes it clear to understand the current challenges and prior work, as well as the proposed methodology and its effectiveness, along with the contributions.

Significance:

The claimed significance of the proposed methodology is convincing. Even though the improvement in absolute scores under the presented benchmarks is not relatively high (but it is non-trivial), the method does address (to a degree, as the authors claim) a considerable and non-negligible issue (rotational misalignment) that is present in the existing approaches for federated learning via LoRA.

Originality:

The submitted work is novel in terms of the studied gaps and proposed solutions in the domain of federated finetuning of large language models via LoRA.

---

> ### Author Rebuttal · Authors · 2026-03-31
>
> **Response to W1 (Computational Overhead and Convergence Trade-offs)**
>
> We appreciate the suggestion to analyze the trade-offs between computational overhead and convergence across different ranks. In our setting, however, the additional cost of rotational alignment is negligible, since the SVD is performed on a small $r \times r$ matrix ($r\ll d$). Across all ranks, the per-round overhead is minimal compared to the overall training cost dominated by forward/backward passes. While we do not include explicit runtime measurements in the current submission, the cost of rotational alignment remains lightweight in practice for the ranks considered.
>
> As illustrated in the updated Table 3 (see https://anonymous.4open.science/r/FedRot-LoRA/extra_rank.png), increasing the rank leads to performance saturation rather than performance gains for FedRot-LoRA. As a result, a computation-versus-convergence trade-off curve across varying ranks may not yield actionable insights beyond reinforcing the efficacy of selecting lower ranks. Theoretically, it is plausible that for highly complex tasks, extreme low ranks might underperform because they severely restrict the representational capacity of the adaptation subspace. Nevertheless, we do not observe this capacity bottleneck across any of our evaluated tasks. We will clarify these points in the revision.
>
> **Response to Q1 (Performance at Extreme Ranks)**
>
> We thank the reviewer for this suggestion. To better understand rank scalability, we conducted additional experiments at both lower and higher ranks ($r=2$ and $r=24$), and will include results covering $r \in \lbrace 2, 4, 8, 16, 24\rbrace$ (see https://anonymous.4open.science/r/FedRot-LoRA/extra_rank.png).
>
> We observe that at low rank ($r=2$), FedRot-LoRA maintains competitive performance, indicating that the alignment step does not hinder optimization even under highly constrained capacity. At higher ranks ($r\geq 16$), FedRot-LoRA remains stable and continues to outperform baselines, while competing methods exhibit more noticeable degradation. These results suggest that rotational alignment is effective across a wide range of ranks.
>
> **Response to Q2 (Training Dynamics and Convergence Curves)**
>
> We thank the reviewer for this suggestion. We agree that visualizing training dynamics provides valuable insight into how rotational alignment affects optimization. To address this, we include the accuracy convergence curve on the MNLI dataset in the revision (see https://anonymous.4open.science/r/FedRot-LoRA/acc_curve.png).
>
> We observe that FedRot-LoRA achieves faster and more stable early-stage convergence and reaches a higher, more stable final performance, and that training trajectories are smoother with fewer fluctuations, indicating reduced aggregation noise compared to baselines. These results are consistent with our motivation: aligning client subspaces mitigates destructive interference during aggregation, leading to more stable and efficient optimization. The curve is averaged over 3 random seeds.
>
> **Response to Q3 (Figure 3 Individual Curves)**
>
> We thank the reviewer for this helpful suggestion. We agree that showing individual baseline curves provides a clearer comparison. We have updated Figure 3 to include individual curves for all baselines (see https://anonymous.4open.science/r/FedRot-LoRA/baselines.png), which will replace the original averaged visualization in the revised manuscript.

---

> > ### Author Rebuttal · Reviewer_miTu · 2026-04-06
> >
> > Thank you for addressing my questions and points. I will maintain my current scores.

---

> > > ### Author Response · Authors · 2026-04-06
> > >
> > > Thank you for maintaining your positive score. We appreciate your constructive questions and feedback, and we will incorporate these refinements in the revision.

---

### Official Review · Reviewer_cnSR · 2026-03-13

**Soundness:** 3
**Presentation:** 3
**Significance:** 2
**Originality:** 2
**Overall Recommendation:** 4
**Confidence:** 3

**Summary:**

This work identifies rotational misalignment as the major source of the well-studied aggregation error in federated LoRA. The authors propose FedRot-LoRA, a federated LoRA framework that aligns client updates via orthogonal transformations prior to aggregation.

**Compliance With Llm Reviewing Policy:**

Affirmed.

**Final Justification:**

Author's response has addressed my questions. I decide to keep my score leaning towards acceptance.

**Key Questions For Authors:**

See Weakness.

**Limitations:**

yes

**Strengths And Weaknesses:**

Strength:
1. The proposed approach is well-motivated. The algorithm design is straightforward and easy to follow.
2. The authors demonstrate a provable improvement on the aggregation error using the proposed alignment-based method.

Weakness:
1. My major concern of this work is on the empirical evaluation. It seems this work misses several key baselines including FlexLoRA[1], Fedex-LoRA[2], and LoRA-Fair[3].
2. It would be informative to demonstrate the difference in the aggregation error between naive averaging and FedRot_LoRA in practice.
3. As FedOPT[4] has become a mainstream federated optimization paradigm, a server-side adaptive optimizer might provide additional improvement. Does the proposed FedRot_LoRA naturally adapt to such paradigm?  As FedIT has a natural extension leveraging adaptive optimizers on the sever side to update LoRA factors, it is unclear whether the authors compare with such extension or naive averaging. Can you please clarify?
4. The number of clients in the experiment is relatively small compared to the contemporary literatures.

The final score will be adjusted based on the author's response.

[1] Bai, J., Chen, D., Qian, B., Yao, L. and Li, Y., 2024. Federated fine-tuning of large language models under heterogeneous tasks and client resources. Advances in Neural Information Processing Systems, 37, pp.14457-14483.

[2] Singhal, R., Ponkshe, K. and Vepakomma, P., 2024. Fedex-lora: Exact aggregation for federated and efficient fine-tuning of foundation models. arXiv preprint arXiv:2410.09432.

[3] Bian, J., Wang, L., Zhang, L. and Xu, J., 2025. LoRA-FAIR: Federated LoRA fine-tuning with aggregation and initialization refinement. In Proceedings of the IEEE/CVF International Conference on Computer Vision (pp. 3737-3746).

[4] Reddi, S., Charles, Z., Zaheer, M., Garrett, Z., Rush, K., Konečný, J., Kumar, S. and McMahan, H.B., 2020. Adaptive federated optimization. arXiv preprint arXiv:2003.00295.

---

> ### Author Rebuttal · Authors · 2026-03-31
>
> **Response to W1 (Baseline Selection)**
>
> We thank the reviewer for this helpful suggestion. To strengthen the empirical evaluation, we have added FlexLoRA to all experimental settings.
> - Updated Table 1: https://anonymous.4open.science/r/FedRot-LoRA/main_exp.png
> - Updated Table 3: https://anonymous.4open.science/r/FedRot-LoRA/extra_rank.png
> - Updated Table 4: https://anonymous.4open.science/r/FedRot-LoRA/non_iid.png
> - Updated Table 5: https://anonymous.4open.science/r/FedRot-LoRA/generative.png
>
> Our baseline selection is designed to compare methods under a communication-efficient federated LoRA setting, where only low-rank adapters are transmitted. In this context: FedIT represents naive factor-wise aggregation, FFA-LoRA and RoLoRA mitigate rotational misalignment via parameter-freezing strategies, and FlexLoRA (now included) represents decomposition-based aggregation. Our updated results show that while FlexLoRA can achieve competitive performance on simpler datasets, it incurs substantial instability in more complex settings (more heterogeneity, more clients).
>
> As requested by multiple reviewers, here we summarize why several other methods were not selected for direct comparison. These methods target complementary settings, introduce additional costs, or alter the core learning objective, making them less directly comparable under our defined constraints. Methods such as FedEx-LoRA and FLoRA transmit additional residuals or high-rank information, increasing communication costs beyond standard LoRA. FedSRD primarily targets communication compression through sparsification and reconstruction in the full-weight space, introducing design components beyond aggregation. LoRA-FAIR involves extra server-side optimization, which incurs extra computation and delay. FedSA-LoRA focuses on partial/personalized aggregation rather than learning a unified global model.
>
> These methods are therefore not directly comparable under the setting considered in our work. We will clarify this comparison scope and include a summary of computation and communication differences in the revision: https://anonymous.4open.science/r/FedRot-LoRA/comparison.png.
>
> Notably, FedRot-LoRA introduces only a lightweight client-side alignment step (orthogonal Procrustes), with complexity $O(dr^2+r^3)$, and does not increase communication cost relative to standard LoRA.
>
> **Response to W2 (Aggregation Error)**
>
> We thank the reviewer for this helpful suggestion. We agree that directly measuring aggregation error provides important empirical insight. In our formulation, the aggregation error $E^{t}$ (Eq. 7) captures the discrepancy introduced by naive factor-wise averaging. In Table 2, we already report this quantity and observe that FedRot-LoRA reduces aggregation error by an order of magnitude compared to naive averaging (FedIT) across QQP, RTE, and MNLI. To further strengthen this point, we have extended the evaluation to all GLUE datasets, consistently observing that rotational alignment significantly reduces aggregation discrepancy in practice: https://anonymous.4open.science/r/FedRot-LoRA/agg_error.png.
>
> **Response to W3 (Compatibility with FedOPT)**
>
> We thank the reviewer for this important question. In the current submission, we adopt a standard FedAvg-style server update, where global LoRA factors are obtained via simple averaging of (aligned) client factors. Accordingly, the FedIT baseline corresponds to naive factor-wise averaging without a FedOPT-style server optimizer.
>
> FedRot-LoRA is fully compatible with FedOPT-style adaptive server updates. Our method operates on the client side by aligning local LoRA factors before aggregation, while preserving the underlying update due to the rotational invariance of LoRA (i.e., $BA$ remains unchanged under orthogonal transformations). The server-side averaging step can be directly replaced by any adaptive optimizer applied to the aligned updates.
>
> We view FedRot-LoRA and FedOPT as complementary: FedRot-LoRA reduces aggregation error within each round, while FedOPT improves optimization dynamics across rounds. We expect the two to be naturally combinable and mutually beneficial.
>
> We will clarify these points in the revision.
>
> **Response to W4 (Number of Clients)**
>
> We thank the reviewer for this observation. To address this, we have extended our experiments to a larger federated setting with 50 clients on GLUE tasks (see https://anonymous.4open.science/r/FedRot-LoRA/main_exp.png). The results show that as the number of clients increases, overall performance degrades for all methods due to stronger heterogeneity and more challenging aggregation. Importantly, the performance gap between FedRot-LoRA and baselines further widens in this regime, highlighting its robustness under large-scale federated settings.
>
> We will include these additional results in the revision to better demonstrate scalability with respect to the number of clients.

---

> > ### Author Rebuttal · Reviewer_cnSR · 2026-04-03
> >
> > I thank the reviewer for the detailed response and the additional experiments. I have several follow-up questions --
> >
> > W1: It's surprising to see FlexLoRA falls short in some experimental settings. The SVD step of FlexLoRA guarantees that the aggrgation error is minimized. What's the intuitive explanation of FedRot-LoRA's better performance? Does FedRot-LoRA has a secret sauce that is distinct from the paper's motivation on reducing aggregation error, or is it just because of insufficient tuning on hyper-parameters?
> >
> > W3: i appreciate the clarification on the compatibility of FedRot-LoRA and FedOPT paradigm. However, as FedOPT applies adaptive optimizers on the server side which explicitly changes the directions of the gradients/updates, I may doubt whether aligning the directions of client updates **before** the adaptive step is still meaningful enough or yield sufficient performance improvement. While this can be a future work, I think the true significance of the proposed method can only be fully validated through empirical evaluation using the state-of-the-art federated learning paradigms (FedOPT).

---

> > > ### Author Response · Authors · 2026-04-05
> > >
> > > **Response to follow-up on W1**
> > >
> > > We thank the reviewer for this insightful follow-up. A key point is that FlexLoRA and FedRot-LoRA optimize **different objectives**. FlexLoRA first forms the ideal full-space aggregate $W_{ideal}$, then computes its best rank-$r$ approximation via truncated SVD. This minimizes reconstruction error to $W_{ideal}$ in the Frobenius norm, but it is **not equivalent to minimizing the aggregation error** $\lVert E^t\rVert$ defined in our paper.
> > >
> > > This distinction matters under heterogeneity. Although each client update $B_i A_i$ has rank at most $r$, the averaged full-space update $W_{ideal} = \frac{1}{N} \sum_{i=1}^N B_i A_i$ can have effective rank larger than $r$ because it combines diverse client-specific directions. FlexLoRA then projects this update back to rank $r$, which necessarily discards components beyond the top $r$ singular directions. Under stronger non-IID heterogeneity or larger client populations, these discarded directions can become more significant, even if they are not dominant in the global spectrum. FedRot-LoRA reduces misalignment before aggregation at the factor level. This does not eliminate aggregation error entirely, but it removes the spurious mismatch caused by rotational non-identifiability and yields a cleaner low-rank aggregate without post-hoc projection. Empirically, this leads to lower aggregation error (see Table 2 at https://anonymous.4open.science/r/FedRot-LoRA/agg_error.png) and better downstream performance than FlexLoRA in all settings.
> > >
> > > We will include this intuition in the revision to better explain the observed gap between the two methods.
> > >
> > > **Response to follow-up on W3**
> > >
> > > We thank the reviewer for this important follow-up. To address this concern directly, we have additionally evaluated FedIT (without alignment) and FedRot-LoRA (with alignment) under a **FedOPT setting** (see https://anonymous.4open.science/r/FedRot-LoRA/fedopt_style.png) using Adam as the adaptive server optimizer, on 5 GLUE tasks under the same 3-client setup as Table 1. The results show that FedIT collapses on all 5 datasets, whereas FedRot-LoRA achieves performance comparable to the FedAvg-style results reported in the main paper. This demonstrates that rotational alignment remains effective under FedOPT-style training.
> > >
> > > *Why FedRot-LoRA remains effective under FedOPT-style training:* FedOPT applies an adaptive optimizer to the aggregated update (model difference) at the server, whereas FedRot-LoRA improves the quality of this aggregated update before it is passed to the optimizer. If the factors $(B_i^t , A_i^t)$ from different clients are misaligned, then the induced model differences ($\Delta B_i^t=B_i^{t+1}-\bar{B}^t$, similarly for $\Delta A_i^t$) are also misaligned across clients. As a result, factor-wise aggregation ($\frac{1}{N}\sum_{i=1}^N \Delta B_i^t, \frac{1}{N}\sum_{i=1}^N \Delta A_i^t$) produces a distorted estimate of the shared update direction, where components may partially cancel out due to rotational mismatch across client subspaces. While adaptive server optimizers can stabilize training via rescaling and modifying update directions (e.g., through momentum), these operations are applied after aggregation and cannot correct directional bias inherent in their input (i.e., the aggregated update without rotational alignment), leading to unstable training as observed for FedIT. FedRot-LoRA mitigates the rotational noise before computing the model differences ($\Delta B_i^t=\tilde{B}_i^{t+1}-\bar{B}^t$), ensuring aligned update directions across clients and providing a higher-quality input to the server optimizer.
> > >
> > > Therefore, under FedOPT, pre-aggregation alignment remains meaningful rather than redundant. Our additional FedOPT results support this claim empirically. We will include these results and clarify this discussion in the revision.

---

### Official Review · Reviewer_mmU5 · 2026-03-13

**Soundness:** 3
**Presentation:** 4
**Significance:** 3
**Originality:** 3
**Overall Recommendation:** 4
**Confidence:** 3

**Summary:**

This paper proposes an algorithm for aggregating low-rank LoRA updates in distributed LoRA training. The paper finds that misalignment between low-rank update matrices $A$ and $B$ across clients can cause destructive interference when naive aggregation is used. It proposes using an orthonormal matrix multiplication to align updates across clients. To theoretically support the proposed method, convergence analysis is included to upper bound the gradient norm in terms of the error incurred in the aggregation step. Next the paper argues that the proposed algorithm reduces this error.

**Compliance With Llm Reviewing Policy:**

Affirmed.

**Final Justification:**

I thank the authors for the thoughtful interaction during the review period. After discussion regarding Assumption 4.5, I think it can be reasonably claimed that the theoretical results indicate that the proposed method can improve performance (although the improvement may be very small). The empirical results show that the improvement is significant for some tasks. I will maintain my positive score.

**Key Questions For Authors:**

* The theoretical argument depends on Assumption 4.5. This seems to be a reasonable assumption for a very small $c_0$, but if $c_0$ is too small, the $\lambda$ range in Corollary 4.9 becomes trivial. Could justification be provided (either theoretically or empirically) that Assumption 4.5 holds for a $c_0$ sufficiently large so that the range of $\lambda$ in Corollary 4.9 is reasonably large?
* Have the authors considered using a scheduler for the interpolation factor $\lambda$? This factor is motivated by noise in early stages of training. Would results improve if a $\lambda$ close to zero were used at the beginning of training and if $\lambda$ were gradually increased to one as training progresses?
* FlexLoRA and FedSRD, which perform aggregation in full-weight space, are described in the Related Works but not included in the experiments. It is reasonable to say that these should not be directly compared to FedRot-LoRA because they are more computationally expensive, but performance comparison would still be informative for users making decisions about trade-offs between computational expense and performance. Was there a more specific reason for not providing these results?

**Limitations:**

Yes

**Strengths And Weaknesses:**

**Strengths**
* The introduction section gives clear and persuasive motivation for the proposed algorithm. The problem the identified, aggregation error, strongly motivates the proposed rotational alignment.
* The proposed method is supported both theoretically with convergence proofs and empirically with benchmark results. The theoretical analysis shows the important role that misalignment error plays in convergence. The empirical results show that their method maintains or slightly improves performance on easy benchmarks while improving it more substantially on difficult benchmarks.
* Tables 3 and 4 help answer potential questions about the generality of the proposed method by showing how different problem settings affect performance relative to baseline methods. This helps improve the overall persuasiveness of the paper.

**Weaknesses**
* Assumption 4.5 presupposes that the alignment gain $\alpha(\lambda)$ of the proposed algorithm is positive and nontrivial, but this is an important point that should be demonstrated, not assumed. There is no a priori reason to think that the lower bound on the alignment gain should be linear in $\lambda$. More importantly,  if the constant $c_0$ is too small, the valid $\lambda$ range in Corollary 4.9 may be very small.
* Best case evaluation results are shown for a grid search over learning rate and alignment strength parameter $\lambda$ for FedRot-LoRA. For baseline methods, grid search is only performed over learning rate. Hence more hyperparameter tuning options are used for FedRot-LoRA than for the baseline methods. This limits the persuasiveness of the empirical results.

---

> ### Author Rebuttal · Authors · 2026-03-31
>
> **Response to W1 & Q1 (Assumption 4.5 and Alignment Gain)**
>
> We thank the reviewer for this important observation. Assumption 4.5 is not intended to suggest that the alignment gain $\alpha(\lambda)$ is globally linear, but rather to impose a first-order lower envelope near $\lambda = 0$. Since $\alpha(\lambda) = 1-\Phi(\lambda)/\Phi(0)$ with $\alpha(0)=0$, the condition $\alpha(\lambda)\ge c_0\lambda$ captures the existence of a positive first-order reduction in dispersion when moving toward alignment. This is a standard regularity condition used to obtain a tractable bound, and our analysis does not rely on global linearity of $\alpha(\lambda)$.
>
> The requirement $c_0>0$ is mild in the heterogeneous settings. When client factors are misaligned (i.e., $R^*\neq I$), moving toward the aligned direction reduces dispersion, implying a positive first-order gain. The degenerate case $c_0=0$ corresponds to already aligned factors, where no improvement from alignment is expected.
>
> Regarding the concern that a small $c_0$​ may restrict the feasible range in Corollary 4.9, we emphasize that the corollary provides a conservative sufficient condition under worst-case constants rather than a tight characterization of all effective $\lambda$. In practice, we observe that the effective range of $\lambda$ is not narrow (Figure 3), where a broad interval yields near-optimal performance. We will clarify this point and the role of the assumption in the revision.
>
> **Response to W2 (Hyperparameter Tuning Fairness)**
>
> We thank the reviewer for this important point. While FedRot-LoRA introduces an additional hyperparameter $\lambda$, we find that performance is not sensitive to its precise value. As shown in Fig. 3(a), a broad range of values (e.g., $\lambda\in\lbrace 0.3,0.5,0.6,0.7\rbrace$) achieves performance comparable to the best setting $\lambda=0.4$. To further support this, we include additional sensitivity plots across GLUE datasets (see https://anonymous.4open.science/r/FedRot-LoRA/lambda_sensitivity.png), which consistently show that near-optimal performance is obtained over a wide range of $\lambda$, which indicates that no fine-grained tuning is needed in practice; instead, a fixed default (e.g., $\lambda = 0.4$) or a very small coarse search over a few values is sufficient. We will clarify this point in the revision.
>
> **Response to Q2 (Adaptive $\lambda$ Scheduling)**
>
> We thank the reviewer for this insightful suggestion. We agree that using a scheduler for $\lambda$ (starting from a smaller value to mitigate early-stage noise and gradually increasing it) is well-motivated and consistent with our design of soft alignment. In our current experiment, we adopt a fixed $\lambda$ for simplicity and to avoid introducing additional scheduling hyperparameters. As shown in Fig. 3, a moderate $\lambda$ already provides stable and strong performance across training stages, suggesting that explicit scheduling is not strictly necessary. That said, adaptive scheduling is a promising extension that could further improve robustness, particularly in early rounds. We will discuss this direction in the revision.
>
> **Response to Q3 (Baselines: FlexLoRA and FedSRD)**
>
> We thank the reviewer for this valuable suggestion. We agree that comparing against full-weight aggregation methods is informative for understanding the trade-off between performance and cost. To this end, we have added FlexLoRA to all experimental settings (see https://anonymous.4open.science/r/FedRot-LoRA/main_exp.png). Regarding FedSRD, while it also operates in the full-weight space, it introduces additional sparsification and reconstruction components and requires a more complex communication pipeline. As a result, its performance depends not only on aggregation but also on compression and reconstruction design choices, making direct comparison less controlled and more challenging. That said, we agree that such comparisons can be informative from a system perspective. We will clarify this distinction in the revision and discuss the trade-offs between lightweight alignment (as in FedRot-LoRA) and full-weight aggregation methods.

---

> > ### Author Rebuttal · Reviewer_mmU5 · 2026-04-03
> >
> > I thank the authors for the thoughtful response. The point that performance is insensitive to selection of alignment parameter $\lambda$ is a helpful clarification that resolves my concerns regarding Weakness 2. I still have some concerns regarding Weakness 1 and Question 1. I agree that the empirical results indicate that $c_0$ is not trivially small and that the effective range of $\lambda$ is not narrow. However, if the paper claims that the proposed method is both empirically and theoretically supported, the theoretical results should not depend on downstream performance results. I think the soundness of the paper would be improved by some justification for Assumption 4.5. Overall, I think this paper is a good contribution, and I will maintain my positive score.

---

> > > ### Author Response · Authors · 2026-04-04
> > >
> > > We thank the reviewer for the helpful clarification and for maintaining the positive score. We agree that the theoretical results should be interpreted independently of downstream empirical performance, and we will revise the paper to make this point clearer.
> > >
> > > Our theoretical contribution is a conservative guarantee: under Assumption 4.5, Theorem 4.8 and Corollary 4.9 show that there exists a non-trivial range of $\lambda$ for which rotational alignment yields a strictly tighter aggregation-error upper bound than naive factor-wise averaging. In other words, the theory is intended to establish the existence of a favorable regime, rather than to fully explain the entire empirical behavior of FedRot-LoRA.
> > >
> > > We agree that Assumption 4.5 deserves clearer justification. In the revision, we will clarify that Assumption 4.5 should be viewed as a regularity condition for the non-degenerate misaligned regime. The empirical results play a complementary role: not to justify the theorem itself, but to show that the favorable range of $\lambda$ is practically common and not narrow.

---

### Official Review · Reviewer_9B9F · 2026-03-14

**Soundness:** 2
**Presentation:** 3
**Significance:** 2
**Originality:** 3
**Overall Recommendation:** 3
**Confidence:** 4

**Summary:**

This paper proposes a framework FedRot-LoRA to address the aggregation error in federated LoRA fine-tuning caused by the misalignment of local update subspaces. The method aligns LoRA factors by solving the orthogonal Procrustes problem before clients upload their parameters to the server. It also introduces an interpolation-based soft rotation mechanism to alleviate the problem of training instability at the early stages.

**Compliance With Llm Reviewing Policy:**

Affirmed.

**Final Justification:**

I still have concerns regarding the significance of improvement and the cost, so I keep my original score.

**Key Questions For Authors:**

1.	The theoretical result on the tighter bound depends critically on Assumptions 4.5-4.7. It would be great to verify whether these assumptions are still consistent with observed training behavior. Is there any possibility that the authors could provide additional experiments or visualizations that verify these assumptions?

2.	The appendix shows that the optimal coefficient varies largely, depending on the task and the degree of data heterogeneity. For example, ranging from 0.2 on GSM8K to 0.8 on the IID MNLI task. How to select or tune such a sensitive hyperparameter in a new federated task with an unknown data distribution?

**Limitations:**

Yes

**Strengths And Weaknesses:**

Strengths:

1.	The paper provides a convergence analysis for standard non-convex optimization in Theorem 4.4. In addition, Theorem 4.8 also provides a mathematical proof that rotational alignment can achieve a tighter upper bound for aggregation error, compared to naive factor-wise averaging. This proof and analysis integration is uncommon in current FL research.

2.	Unlike methods that need to transmit high-rank residuals, such as FedEx-LoRA, or aggregating in the full parameter space, such as FlexLoRA, FedRot-LoRA strictly maintains the low-rank communication constraints of LoRA without increasing any communication overhead.

3.	The extensive experimental results on different tasks and good ablations demonstrate the effectiveness of the proposed method. Algorithms and Equations are also clearly presented, with an emphasis on reproducibility.

Weakness:

1.	However, the empirical results do not fully justify the strong claim that FedRot-LoRA consistently outperforms current federated LoRA baselines. On the GLUE benchmark, the average improvement over the strongest baseline RoLoRA is modest, rising from 0.8862 to 0.8932 for N=3 and from 0.8786 to 0.8818 for N=10. On RTE with N=10, FedRot-LoRA actually performs slightly worse than RoLoRA (0.798 versus 0.805). The gains over FFA-LoRA on generation tasks are likewise inconsistent. On GSM8K, for example, the improvement is limited, increasing only from 0.4361 to 0.4437.

2.	The baseline comparisons also appear selective. Although the related work discusses several relevant methods, including FedSA-LoRA, FlexLoRA, FedSRD, FedEx-LoRA, and FLoRA, the main experiments compare against only FedIT, FFA-LoRA, and RoLoRA. The authors claimed that these omitted methods incur high computational or communication overhead, but this statement requires experiments to validate. For a paper that considers itself as superior without increasing any communication costs, the lack of reported communication statistics is a clear limitation.

3.	The authors claim that their approach still works well across different client scales, LoRA ranks, and levels of heterogeneity. However, the actual experimental scope presented in the paper is considerably narrower. For example, the GLUE evaluation is only tested on 3 and 10 clients, while studies on rank and heterogeneity focus only on the MNLI task, and GSM8K task even uses an IID setting. As a result, the paper's claims are only partially supported by the experiments.

---

> ### Author Rebuttal · Authors · 2026-03-31
>
> **Response to W1 & W3 (Empirical Improvements and Experimental Scope)**
>
> We thank the reviewer for these observations. As the concerns about improvement magnitude (W1) and evaluation scope (W3) are closely related, we address them jointly below.
>
> We clarify that our claim of "consistent improvement" refers to robust gains across challenging federated regimes, rather than uniformly large margins in all settings. While improvements on relatively well-aligned or low-complexity settings (e.g., small client counts or low rank) can be modest, the advantage of FedRot-LoRA becomes substantially more pronounced when aggregation is intrinsically difficult, such as under high rank, strong heterogeneity, or many clients. In particular, under high-rank settings where rotational misalignment is amplified, FedRot-LoRA yields large gains. For example, at r=16 (Table 3), RoLoRA achieves 0.722, whereas FedRot-LoRA reaches 0.865 (+14.3 points). This reflects the fact that higher-rank LoRA introduces a more complex latent subspace, where misalignment across clients significantly degrades factor-wise averaging. By explicitly aligning subspaces prior to aggregation, FedRot-LoRA mitigates this destructive interference.
>
> We acknowledge that the original submission evaluated rank, heterogeneity, and client scale in a limited scope. To strengthen this point, we have expanded our evaluation along three axes:
> - Client scale: up to 50 clients: https://anonymous.4open.science/r/FedRot-LoRA/main_exp.png
> - Rank scaling across tasks: added SST-2 and QNLI
>   - SST-2: https://anonymous.4open.science/r/FedRot-LoRA/sst2_rank.png
>   - QNLI: https://anonymous.4open.science/r/FedRot-LoRA/qnli_rank.png
> - Data heterogeneity across tasks: added SST-2 and QNLI
>   - SST-2: https://anonymous.4open.science/r/FedRot-LoRA/sst2_noniid.png
>   - QNLI: https://anonymous.4open.science/r/FedRot-LoRA/qnli_noniid.png
>
> These additional results will be included in the revision.
>
> For GSM8K, we follow prior federated LLM work and adopt an IID split due to the lack of natural client partitions \[1,2,3\]. In this setting, where aggregation is less challenging, improvements are correspondingly smaller. Nevertheless, FedRot-LoRA remains competitive and shows stronger gains on more complex generative tasks such as HumanEval.
>
> Overall, these results suggest that FedRot-LoRA provides robust improvements in precisely the regimes where aggregation error becomes a dominant bottleneck, which is the primary focus of our work.
>
> \[1\] Kuang, Weirui, et al. FederatedScope-LLM: A Comprehensive Package for Fine-tuning Large Language Models in Federated Learning.
>
> ​​\[2\] Guo, Pengxin, et al. Selective aggregation for low-rank adaptation in federated learning.
>
> \[3\] Huang, Jiayu, et al. Stabilized Fine-Tuning with LoRA in Federated Learning: Mitigating the Side Effect of Client Size and Rank via the Scaling Factor.
>
> **Response to W2 (Baselines and Communication Cost)**
>
> We thank the reviewer for this important point, also raised by Reviewer cnSR; please see our response to cnSR's W1 (not repeated here for brevity).
>
> **Response to Q1 (Verification of Theoretical Assumptions)**
>
> We thank the reviewer for this suggestion. To validate Assumptions 4.5-4.7, we provide empirical measurements of the corresponding quantities during training.
> - Assumption 4.5 (positive alignment gain): we track $\alpha(\lambda)$ for $\lambda\in\lbrace 0.2, 0.4, 0.6\rbrace$ during training and observe that it remains positive (see https://anonymous.4open.science/r/FedRot-LoRA/gain_alpha.png).
> - Assumption 4.6 (client drift): we track $\lVert B_i^t - B_{ref} \rVert_F$ and $\lVert A_i^t - A_{ref} \rVert_F$, and both remain bounded away from zero across training, indicating persistent client drift (see https://anonymous.4open.science/r/FedRot-LoRA/drift.png).
> - Assumption 4.7 (bounded dispersion): we track $\lVert R_i^{t,*} - I\| / \|A_i^t - A_{ref}\rVert_F$, which remains bounded during training, supporting the existence of a finite constant $\kappa$ (see https://anonymous.4open.science/r/FedRot-LoRA/kappa.png).
>
> Overall, these plots show that the assumptions are consistent with observed training behavior in heterogeneous federated settings rather than pathological cases.
>
> **Response to Q2 (Hyperparameter Sensitivity of $\lambda$)**
>
> While the optimal $\lambda$ varies across tasks and heterogeneity levels, our results show that performance is not sharply sensitive within a moderate range. For example, in Fig. 3(a), $\lambda \in \lbrace 0.3, 0.5, 0.6, 0.7 \rbrace$ performs comparably to the best value $\lambda=0.4$. We will include additional sensitivity plots showing that a broad interval of $\lambda$ yields near-optimal performance across settings: https://anonymous.4open.science/r/FedRot-LoRA/lambda_sensitivity.png. In practice, $\lambda$ does not require expensive tuning: a moderate default (e.g., $\lambda=0.4$) or a small grid over 2-3 nearby values with a short pilot run is sufficient.

---

> > ### Author Rebuttal · Reviewer_9B9F · 2026-04-04
> >
> > Thank you very much for providing answers to some of my concerns and questions. However, I am still not fully convinced on some points after reading the rebuttal and please see details below:
> >
> > 1. I respectively not fully agree with the author's claim of "robust gains across challenging federated regimes". E.g., for the On RTE with N=10, FedRot-LoRA actually performs slightly worse than RoLoRA (0.798 versus 0.805). In addition, some of the gains are pretty thin (less than 0.5%), I am not sure whether such gains are meaningful enough to be useful in practice.
> >
> > 2. For Baselines and Communication Cost, I also share similar concerns raised in cnSR's comments.
> >
> > 3. I am not fully convinced by the answers on Hyperparameter Sensitivity. The authors provided a plot and saying a moderate default (e.g., $\lambda=0.4$) works well. However, they did not show GSM8K results on the plot and the paper uses 0.2 as hyperparameter value on GSM8K.

---

> > > ### Author Response · Authors · 2026-04-05
> > >
> > > **Response to follow-up on empirical gains and robustness**
> > >
> > > We thank the reviewer for the careful follow-up. We agree that the original wording “robust gains across challenging federated regimes” may be too strong if interpreted as uniform improvement in every setting, and we will revise the paper to state this more precisely.
> > >
> > > For the RTE ($N=10$), the reported difference is $0.798 \pm 0.005$ (FedRot-LoRA) vs. $0.805 \pm 0.007$ (RoLoRA). The gap ($0.007$) is comparable to the run-to-run standard deviation, so we view these two methods as performing similarly in this setting rather than as evidence of a meaningful degradation.
> > >
> > > More importantly, our additional experiments at a larger **client scale** show that the advantage of FedRot-LoRA becomes substantially clearer when aggregation is more difficult. In the 50-client setting (https://anonymous.4open.science/r/FedRot-LoRA/main_exp.png), the average improvement over baselines increases markedly across the 5 GLUE tasks. This is the regime we intend to emphasize: FedRot-LoRA is most beneficial when heterogeneity, client count, or rank make factor aggregation genuinely difficult.
> > >
> > > We will therefore revise the paper to avoid over-general wording and instead state that the gains are **most pronounced in harder federated regimes**, while remaining competitive in easier settings.
> > >
> > > **Response to follow-up on baselines, communication cost, and FlexLoRA**
> > >
> > > We thank the reviewer for this follow-up. We have added FlexLoRA to the experiments and will include both its performance and a summary of communication/computation costs in the revision.
> > >
> > > Regarding the intuition: FlexLoRA and FedRot-LoRA optimize **different objectives**. FlexLoRA performs aggregation in the full-weight space ($\frac{1}{N}\sum_{i=1}^N B_i^t A_i^t$) and then computes a best rank-$r$ approximation via truncated SVD. This minimizes reconstruction error to the full-space aggregate, but it does not directly address factor-space misalignment before aggregation.
> > >
> > > Under stronger heterogeneity or larger client populations, the ideal full-space aggregate can have effective rank larger than $r$. In this case, the truncated SVD step necessarily discards lower-energy directions. Some of these directions may still be important for subsets of clients, even if they do not dominate the global singular spectrum. This makes the resulting low-rank global adapter a weaker initialization for subsequent rounds. By contrast, FedRot-LoRA reduces the mismatch before aggregation at the factor level, producing a cleaner low-rank update without requiring full-space reconstruction and projection. This is the main reason we observe better performance than FlexLoRA in the experiments.
> > >
> > > We also clarify in the revision that other omitted methods (e.g., FedEx-LoRA, FLoRA) are not directly comparable under the same communication budget because they transmit additional information beyond standard low-rank adapters.
> > >
> > > **Response to hyperparameter sensitivity (including GSM8K)**
> > >
> > > We thank the reviewer for pointing this out, and we apologize for not including GSM8K in the original sensitivity plot. We have now added sensitivity plots for the generative tasks (GSM8K and HumanEval): https://anonymous.4open.science/r/FedRot-LoRA/generative_sensitivity.png.
> > >
> > > These plots show that performance is **not sharply sensitive** to $\lambda$, but we agree that the best value is **task-dependent**. In particular, GSM8K uses an IID partition, where client updates are already relatively well aligned, and in this case a smaller value such as $\lambda = 0.2$ works best. This is consistent with our interpretation that weaker alignment is sufficient when subspace mismatch is limited.
> > >
> > > Accordingly, we will revise the paper to avoid implying a universal default such as $\lambda = 0.4$. A more accurate practical recommendation is:
> > > - use a **small coarse search** over a few discrete values (e.g., $\lambda \in \lbrace 0.2,0.4,0.6\rbrace$ ); or
> > > - select a smaller $\lambda$ in nearly IID settings and a moderate $\lambda$ in more heterogeneous ones.
> > >
> > > We will include the generative-task sensitivity plots and clarify this guidance in the revision.

---

### Decision · Program_Chairs · 2026-04-30

**Decision:**

Accept (regular)

**Comment:**

This paper addresses a critical failure mode in Federated LoRA: rotational misalignment. The authors identify that because low-rank factorizations are rotationally invariant, semantically equivalent local updates can be represented in different latent subspaces across clients. When these factors are averaged directly, they interfere destructively, leading to aggregation error and training instability. To mitigate this, the authors propose FedRot-LoRA, which aligns client updates via orthogonal transformations (Procrustes problem) prior to aggregation. The framework includes an interpolation-based soft rotation mechanism to stabilize early training. The authors have proactively addressed reviewer concerns by adding experiments, clarifying theoretical assumptions, and expanding their baseline comparisons. The consensus among reviewers leaned toward acceptance after the rebuttal phase, while Reviewer 9B9F pointed out the remaining concern over the consistence of improvement and the cost. Given the novelty and contribution, I would recommend a weak acceptance, provided that the authors revise the paper about the improvement claims and add additional evidence on cost in their final manuscript.